# Genomic perspective on the bacillus causing paratyphoid B fever

Jane Hawkey [1,25], Lise Frézal [2,25], Alicia Tran Dien[2,24], Anna Zhukova [3], Derek Brown [4], Marie Anne Chattaway[5], Sandra Simon[6], Hidemasa Izumiya [7], Patricia I. Fields [8], Niall De Lappe[9], Lidia Kaftyreva[10], Xuebin Xu[11], Junko Isobe[12], Dominique Clermont [13], Elisabeth Njamkepo [2], Yukihiro Akeda[7], Sylvie Issenhuth-Jeanjean[2], Mariia Makarova [10], Yanan Wang [14,15], Martin Hunt[16,17,18,19], Brent M. Jenkins [8], Magali Ravel[2], Véronique Guibert[2], Estelle Serre[2], Zoya Matveeva[10], Laëtitia Fabre[2], Martin Cormican [9,20], Min Yue[21,22], Baoli Zhu [15], Masatomo Morita [7], Zamin Iqbal [16,23], Carolina Silva Nodari[2], Maria Pardos de la Gandara [2] & François-Xavier Weill [2] ✉

Paratyphoid B fever (PTB) is caused by an invasive lineage (phylogroup 1, PG1) of *Salmonella enterica* serotype Paratyphi B (SPB). However, little was known about the global population structure, geographic distribution, and evolution of this pathogen. Here, we report a whole-genome analysis of 568 historical and contemporary SPB PG1 isolates, obtained globally, between 1898 and 2021. We show that this pathogen existed in the 13th century, subsequently diversifying into 11 lineages and 38 genotypes with strong phylogeographic patterns. Following its discovery in 1896, it circulated across Europe until the 1970s, after which it was mostly reimported into Europe from South America, the Middle East, South Asia, and North Africa. Antimicrobial resistance recently emerged in various genotypes of SPB PG1, mostly through mutations of the quinolone-resistance-determining regions of *gyrA* and *gyrB*. This study provides an unprecedented insight into SPB PG1 and essential genomic tools for identifying and tracking this pathogen, thereby facilitating the global genomic surveillance of PTB.

At the turn of the 20th century, investigators in Europe and North America showed that Eberth's bacillus (now known as *Salmonella enterica* serotype Typhi) was not the only organism causing enteric fever, a severe systemic infection causing prolonged high fever, fatigue, headache, and abdominal pain, exclusively in humans[1–3]. The other causal bacteria identified were paratyphoid bacilli of three different types: A, B, and C. The first two cases involving the paratyphoid B bacillus (now known as *S. enterica* serotype Paratyphi B and referred to hereafter as SPB) were described by Achard and Bensaude in Paris, France, in 1896 (refs. 1,4,5). Following the introduction of appropriate laboratory tools − initially based on O- and H-antigen serotyping[6]

(recognising the antigenic formula 1,4[5],12:b:1,2) and then on an absence of *d*-tartrate (*d*-Tar⁻) fermentation by the cultured bacteria (SPB⁻ strains)[7] − SPB was more frequently detected in Europe (Supplementary Note "Epidemiology of paratyphoid B fever during the first half of the 20th century"). PTB disease was milder than typhoid fever, with a lower incidence of complications and a lower mortality[8,9]. It occurred as sporadic cases with occasional outbreaks, rarely due to case-to-case transmission in institutions (such as garrisons, hospitals, and children's homes), but was mostly foodborne, particularly in natural or synthetic cream, unpasteurised milk, and bakery products[9]. PTB cases remained frequent across Europe in the 1950s and 1960s[10–12], but

the end of the 1970s[13] saw a progressive decline in SPB⁻ isolation accompanied by an increase in the isolation of zoonotic *d*-tartrate-fermenting (SPB⁺) strains, also known as variant Java strains. Epidemiologically, PTB also shifted from being locally acquired to being an imported disease. In the UK, 43.2% (152/352) of PTB cases were considered to have been contracted locally between 1973 and 1977, whereas 56.8% (200/352) were considered to have been contracted abroad, particularly in Mediterranean and Middle Eastern countries[14].

A phage typing scheme was developed for the surveillance of PTB as early as World War II (WWII)[15,16]. This scheme subtyped SPB⁻ isolates into 10 different phage types (PTs) – 1, 2, 3a, 3aI, 3b, BAOR (British Army of the Rhine), Jersey, Beccles, Dundee, and Taunton – and reports based on tens of thousands of isolates phage-typed across the world were regularly published from the 1940s to the 1990s[10–12,17,18]. A particular geographic distribution was observed, with PTs 1 and 2 reported to be autochthonous to the UK (whereas Taunton was found in imported cases); BAOR was prevalent in Central Europe; Dundee was prevalent in France, and the PT 3 series (3a, 3aI, and 3b) was more common in North America than elsewhere. However, despite their different epidemiological characteristics and pathogenicity, SPB⁻ and SPB⁺ strains were grouped together as a single serotype, SPB[6,19], and their PTs were ultimately combined without distinction in world surveys, rendering these surveys less informative.

From the 1990s onwards, population genetics tools were used to distinguish between SPB⁺ and invasive SPB⁻ strains, initially by multilocus enzyme electrophoresis[20] (Pb1 group for SPB⁻) and then by multilocus sequence typing (MLST)[21] (sequence type ST86 and five single-locus variants [SLVs] of ST86 for SPB⁻). In 2003, the molecular basis of the *d*-Tar⁻ character of SPB⁻ strains was elucidated: a single-nucleotide variant (SNV) leading to the loss of the start codon of a gene involved in the *d*-tartrate fermentation pathway[22].

In 2017, an estimated 10·9 million cases of typhoid fever and 3·4 million cases of paratyphoid fever (all types), resulted in 116·8 and 19·1 thousand deaths worldwide, respectively[23]. However, by contrast to *S. enterica* serotypes Typhi (STY)[24], Paratyphi A (SPA)[25], and Paratyphi C[26], little is known about the population structure of SPB⁻. Connor and coworkers[27] were the first to try to elucidate the structure in a study of 191 isolates of SPB (SPB⁺ and SPB⁻) from around the world. Their phylogenomic analysis identified 10 distinct lineages, named phylogroups (PGs 1-10). All 34 SPB⁻ isolates were grouped into the invasive PG1 lineage, which was derived from the closely related lineages PG2 to −5, containing SPB⁺ gastrointestinal isolates. Microbial genomic surveillance conducted in the UK since 2014 led to the identification of phylogenetic clades of SPB⁻ strains associated with travel to South America, Iraq, and Pakistan[28,29]. However, the use of a limited number of global SPB⁻ isolates or of contemporary routine SPB⁻ isolates ruled out any deeper global phylogenomic analysis of this pathogen.

Here, we studied 568 genomes from a spatially and temporally diverse set of SPB⁻ isolates, to determine the global population structure, geographic distribution, and evolution of this pathogen. We also developed a hierarchical SNV-based genotyping scheme implemented within Mykrobe open-source software that splits SPB⁻ isolates into 38 distinct and often phylogeographically informative genotypes, thereby facilitating the global genomic surveillance of PTB.

## Results

### Phylogenomics of *S. enterica* serotype Paratyphi B PG1

We assembled a set of 568 genomes (the "diversity dataset"), including 446 generated specifically for this study, from the widest possible temporal and geographic distribution of available SPB⁻ isolates. These isolates originated from different sources (humans, environment, food, animals), geographic areas (41 countries spanning four continents) and were collected between 1898 and 2021. The number of historical isolates was significant, with 41% (233/568) collected between 1898 and 1980 (Fig. 1, Supplementary Data 1). We ensured that

this diversity dataset comprised exclusively genomes (i) with the correct in silico serotype, (ii) containing the specific SNV associated with the loss of *d*-Tar fermentation in SPB⁻ (ref. 22), and (iii) belonging to the invasive lineage, PG1 (ref. 27). This was achieved in a straightforward manner with the HC400_1620 cluster of the EnteroBase *Salmonella* core-genome MLST scheme (https://enterobase.warwick.ac.uk/species/index/senterica) used as a proxy (Supplementary Note "Validation of the SPB⁻ PG1 diversity dataset", Supplementary Fig. 1).

A maximum likelihood (ML) phylogeny of these 568 SPB⁻ PG1 genomes was constructed from 15,995 single-nucleotide variants (SNVs) distributed over the non-repetitive, non-recombinant core genome (Fig. 1a, Supplementary Fig. 2, Supplementary Data 2). Eleven lineages were identified (L1 to L11), one of which (L10) predominated, accounting for 62% (352/568) of the isolates. The frequency of L10 increased sharply over the study period, from 6.1% (2/33) between 1898 and 1950 to 81.6% (182/223) for the 2001–2021 period (Fig. 1b). L10 was found worldwide, accounting for 38.6% (78/202) of the European isolates (i.e., isolates recovered from infections acquired in Europe) to 88.7% (86/97) of the Middle Eastern isolates (Fig. 1c). Lineage L7 was also widely distributed and was detected in isolates from all geographic regions other than the Middle East (Fig. 1c). Lineage L5 was more frequent in Asia (particularly East Asia), whereas L2 and L9 were more frequently identified in Europe. The only strain (116 K) of serotype Onarimon (an O:9 antigen variant of SPB⁻) − isolated in Japan in 1935 − belonged to lineage L5.

Lineages were further subdivided into well-supported monophyletic groups at various hierarchical levels, including clades, subclades, and an additional higher-resolution group on the basis of both hierBAPS and visual inspection (Fig. 2a). In total, we defined 38 hierarchical genotypes with a phylogenetically informative nomenclature of the form [lineage].[clade].[subclade].[higher-resolution group] (Fig. 2a,b). Strong geographic patterns with differences from country to continent level were observed for 17 genotypes, whereas two genotypes were more widespread, isolated from two continents (Supplementary Data 1). These two genotypes were genotype 7.2 found in Europe (26.3%, 5/19) and Asia (73.7%, 14/19, mostly East Asia), and genotype 10.3.8.5 found in Europe (71.4%, 20/28) and North Africa (25%, 7/28). One genotype, 7.3.2 − also found in Europe (41.2%, 7/17) and North Africa (52.9%, 9/17) − was associated with a particular PT, BAOR (see Comparison of phylogenomics data with other typing schemes) (Supplementary Data 1). This geographic or PT information was added to the genotype nomenclature as an alias, to make it more informative.

The vast majority (*n* = 27, 71%) of the 38 genotypes comprised European isolates. These European isolates were found in the following genotype groups (each containing three or more isolates): 2.0, 2.1, 4, 7.3, 9.0, 9.1_France, 10.1.1_Europe1, 10.2, and 10.3.8.5_EuropeNorthAfrica (Table 1). Our oldest strain (CIP A214) − probably isolated by H. Conradi in Germany in 1898 − belonged to genotype 4. Middle Eastern isolates were found in 50% (19/38) of the genotypes, with particularly high frequencies in genotype 1, 2.1.1_Turkey1, 10.1, 10.3.2_MiddleEast1, 10.3.3_Turkey2, 10.3.5_MiddleEast2, 10.3.8.2_Turkey3, 10.3.8.3_MiddleEast3, and 10.3.8.4_MiddleEast4 groups. American isolates belonged to 34.2% (13/38) of the genotypes and were particularly frequent in the genotype 6, 7.1.1_Chile1, 10.3.4_Chile2, and 10.3.6_SouthAmerica groups. African (excluding Egypt) and Asian (excluding Middle Eastern countries; see the footnotes of Table 1) isolates were each present in 23.7% (9/38) of the genotype groups. The African isolates, which were almost exclusively from North Africa (only three were from East Africa), were most frequently of genotypes 7.3.1_NorthAfrica1, 7.3.2_BAOR, 10.3.7_NorthAfrica2, and 10.3.9_NorthAfrica3. Two East African isolates from Madagascar collected in 1962 and 2001 were of genotype 10.2; whereas a third isolate collected from Ethiopia in 2018 was of genotype 10.0 (Supplementary Data 1). The South Asian isolates belonged exclusively to the genotypes of lineage L10 (in particular

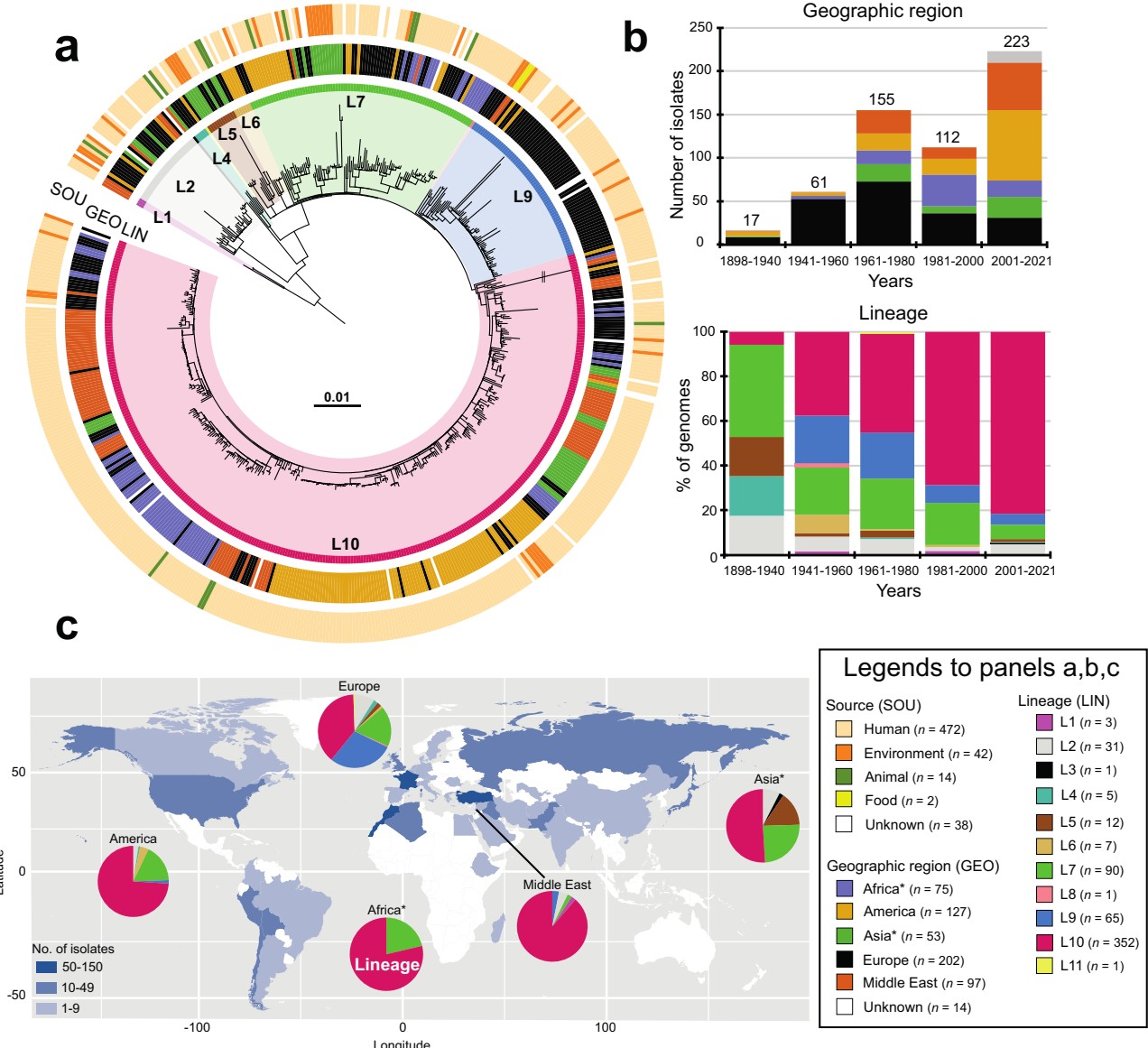

**Fig. 1 | Phylogeny, temporal, geographic, and source distribution of the 568 SPB⁻PG1 isolates from the diversity set (1898 – 2021). a** Circular maximum likelihood phylogeny (rooted on ancestral lineage L1 genome NCTC 8299) for the 568 SPB⁻ PG1 isolates from the diversity dataset. The double slash (//) indicates an artificial shortening of this branch for visualisation. The rings show the associated information for each isolate, according to its position in the phylogeny, from the innermost to outermost, in the following order: (1) lineage (LIN); (2) geographic region (GEO); (3) and source (SOU). Lineages are labelled LX, where X is the lineage number. Lineages L3, L8, and L11, which contain only singletons are not labelled. The tips of the tree are highlighted according to lineage with a lighter hue of the colour used in the innermost ring (LIN). The scale bar indicates the number of

substitutions per variable site (SNVs). **b** The stacked bar chart on the left shows the distribution of the 568 isolates by geographic region and time period, and the stacked bar chart on the right shows the frequencies of the lineages for the same time periods. **c** Number of isolates per country (map) and frequencies of the lineages by world region (pie charts). An asterisk indicates the reassignment of some African and Asian countries to the Middle East (see Table 1). The map was drawn in R with the "ggplot2" package world map data from Wickham H (2016). ggplot2: Elegant Graphics for Data Analysis. Springer-Verlag New York. ISBN 978- 3- 319-24277-4, https://ggplot2.tidyverse.org. Source data are provided as a Source Data file.

10.3.1_SouthAsia1 and 10.3.8_SouthAsia2), whereas almost all the East Asian isolates belonged to genotypes from older lineages (2.1, 5, and 7.2_EuropeEasternAsia). The genotype distribution for animal isolates did not differ significantly from that for human isolates (Supplementary Note "Genotypes found in non-human SPB- PG1 isolates").

After confirming the presence of a strong temporal signal in our dataset (Supplementary Fig. 3), we applied a Bayesian phylogenetic approach to a spatially and temporally representative subset of 256 isolates to estimate the nucleotide substitution rates and divergence times of the different lineages (Table 2) and to construct a dated phylogeny (Fig. 3 and Supplementary Fig. 4). We estimated the

genome-wide substitution rate at $1.2 \times 10^{-7}$ substitutions site$^{-1}$ year$^{-1}$ (95% credible interval (CI) = $9.6 \times 10^{-8}$ to $1.5 \times 10^{-7}$), giving a most recent common ancestor (MRCA) for all SPB⁻ PG1 in our collection dating back to 1274 CE (common era) (95% CI, 915–1583). The MRCAs of the different lineages were estimated to have existed in the 18th century or first half of the 19th century (Table 2).

**Comparison of phylogenomics data with other typing schemes**

World surveys of SPB PTs were regularly reported over several decades following World War II (Supplementary Fig. 5). However, the phylogenetic value of the typing scheme used at the time has never been

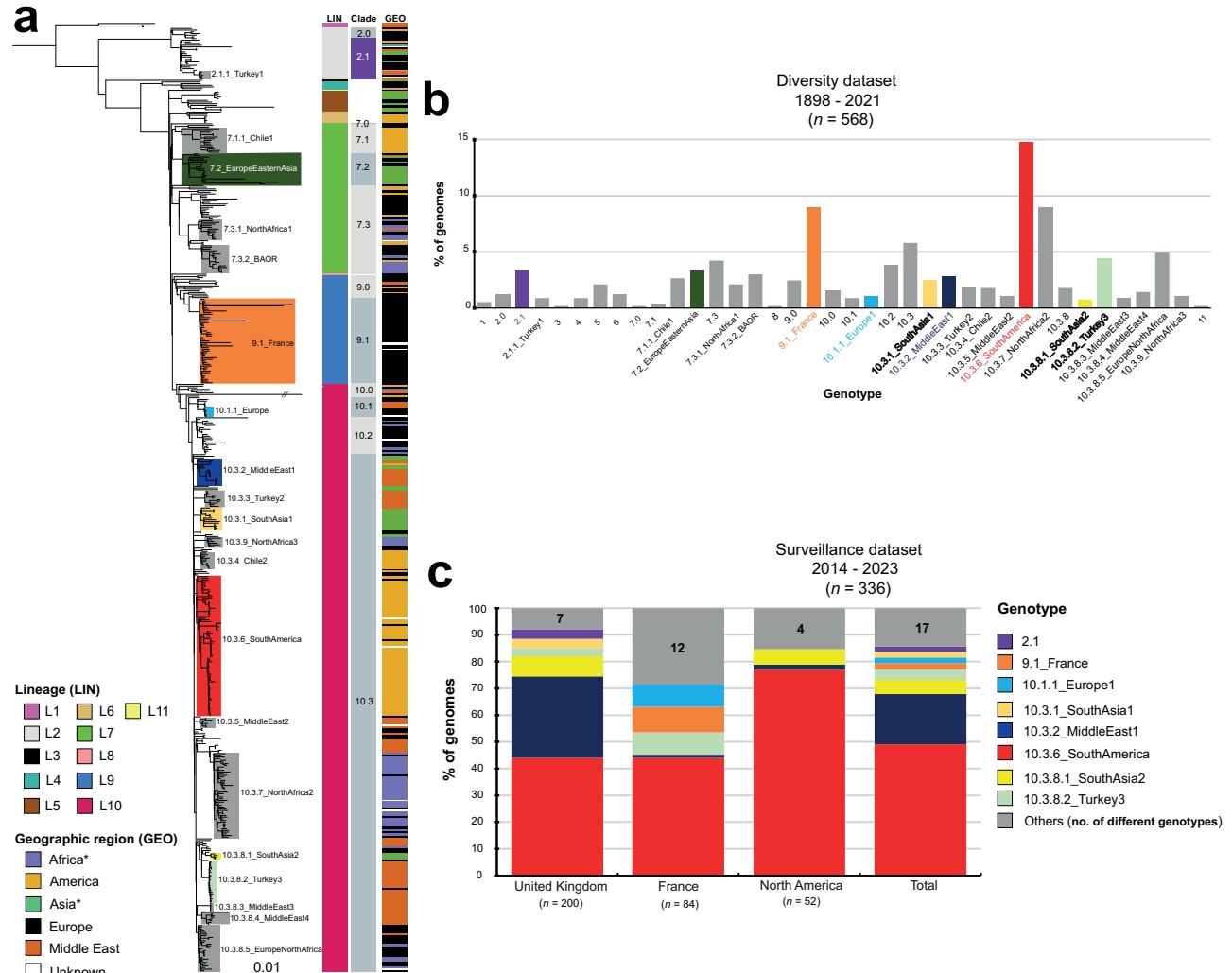

**Fig. 2 | Identification of the 38 hierarchical genotypes of SPB⁻ PG1 and their distribution between the diversity and surveillance datasets. a** Maximum likelihood phylogeny for the 568 SPB⁻ PG1 isolates (as in Fig. 1a, but not circular). The main genotypes are labelled and coloured. Columns on the right indicate the lineage (LIN) (see inset legend), clade and geographic origin (GEO) (see inset legend) of the isolates. An asterisk indicates the reassignment of some African and Asian countries to the Middle East (see Table 1). **b** Frequencies of the 38 genotypes for the 568 genomes of the diversity dataset. The colours are similar to those used in panel "a". **c** Stacked bars indicate the relative abundance of each genotype − coloured as in the legend, inset − for the 336 recent isolates from the UK, France and North America (surveillance dataset). Source data are provided as a Source Data file.

assessed. We, therefore, analysed the correlation between phage-typing and genomic data for the 254 SPB⁻ isolates from the diversity dataset for which phage-typing results were available (Fig. 4). Good concordance was observed for the isolates of four PTs: 92.8% (13/14) of PT 1 isolates belonged to lineage L2, 95.8% (91/95) of those typed as Taunton belonged to L10, and 100% of BAOR (11/11) and Jersey (6/6) isolates belonged to lineage L7 (Supplementary Data 1). The isolates of the BAOR and Jersey PTs even belonged to single genotypes, 7.3.2_BAOR and 7.3, respectively. By contrast, isolates typed as PT Taunton belonged to 10 different genotypes within L10 (Supplementary Data 1). The only PT 2 isolate belonged to L1. The isolates of the remaining PTs were not assigned to a predominant lineage and were instead considered to belong to four lineages. For example, 50.1% (32/63) of the isolates typed as Dundee belonged to L9 (all isolates were from Europe, and all but one belonged to genotype 9.1_France), 46% (29/63) belonged to L10 (from multiple genotypes, mostly corresponding to isolates from the Middle East and North Africa), and 1.6% (2/63) belonged to L2.

Using MLST7, Achtman and coworkers[21] showed that ST86 and five SLVs of ST86 contained only SPB isolates containing the *d*-Tar

-specific SNV. We used this same MLST7 scheme − now implemented in EnteroBase − to analyse the 568 genomes of the diversity dataset. The vast majority of these genomes (96.7%, 549/568) belonged to ST86 (Supplementary Data 1). However, 17 of these genomes belonged to 13 different SLVs of ST86. These 17 isolates included the three lineage L1 genomes, all three belonging to ST5113. One genome (13−80) could not be typed by MLST due to an incomplete *purE* gene. Finally, one genome (ERR129867) described by Connor and coworkers[27] was a triple-locus variant of ST86 (ST2134).

With a higher-resolution cgMLST HC level, such as HC200, it was possible to identify lineages L1 to L4 (L1, HC200_137805; L2, HC200_17706; L3, HC200_301037; L4, HC200_12575) (Supplementary Fig. 6, Supplementary Data 1). However, all other isolates belonging to lineages L5 to L11 were assigned to the same HC200 cluster (HC200_1620), with the exception of three isolates previously found to be outliers in the root-to-tip analysis, which were each assigned to a unique HC200 cluster. With HC100 to HC50, it was not possible to recognise SPB⁻ population structure as determined by our core-genome SNV-based phylogenetic analysis. For example, the isolates belonging to the emerging South American genotype 10.3.6 were

**Table 1 | Geographic distribution of the 38 genotypes found in the diversity dataset**

| Genotype | n | Africa* n (%) | America n (%) | Asia** n (%) | Europe n (%) | Middle East n (%) | Unknown n (%) |
|---|---|---|---|---|---|---|---|
| 1 | 3 | | | | | **3 (100%)** | |
| 2.0 | 7 | | 1 (14.3%) | | **6 (85.7%)** | | |
| 2.1 | 19 | | 2 (10.5%) | 4 (21.1%) | **11 (57.9%)** | 1 (5.3%) | 1 (5.3%) |
| 2.1.1_Turkey1 | 5 | | | | 1 (20.0%) | **3 (60.0%)** | 1 (20.0%) |
| 3 | 1 | | | | **1 (100%)** | | |
| 4 | 5 | | 1 (20.0%) | | **4 (80.0%)** | | |
| 5 | 12 | | | **8 (66.7%)** | 4 (33.3%) | | |
| 6 | 7 | | **5 (71.4%)** | | 2 (28.6%) | | |
| 7.0 | 1 | | | | **1 (100%)** | | |
| 7.1 | 2 | | | | **2 (100%)** | | |
| 7.1.1_Chile1 | 15 | | **15 (100%)** | | | | |
| 7.2_EuropeEasternAsia | 19 | | | **13 (68.4%)** | 5 (26.3%) | 1 (5.3%) | |
| 7.3 | 24 | | 6 (25.0%) | | **17 (70.8%)** | | 1 (4.2%) |
| 7.3.1_NorthAfrica1 | 12 | **7 (58.3%)** | | | 4 (33.3%) | 1 (8.3%) | |
| 7.3.2_BAOR | 17 | **9 (52.9%)** | 1 (5.9%) | | 7 (41.2%) | | |
| 8 | 1 | | | | **1 (100%)** | | |
| 9.0 | 14 | | 2 (14.3%) | | **10 (71.4%)** | 2 (14.3%) | |
| 9.1_France | 51 | | | | **48 (94.1%)** | | 3 (5.9%) |
| 10.0 | 9 | 1 (11.1%) | 2 (22.0%) | | 4 (44.4%) | 2 (22.2%) | |
| 10.1 | 5 | | | | 2 (40.0%) | **3 (60.0%)** | |
| 10.1.1_Europe1 | 6 | | | | **4 (66.7%)** | 1 (16.7%) | 1 (16.7%) |
| 10.2 | 22 | 5 (22.7%) | | | **16 (72.7%)** | | 1 (4.5%) |
| 10.3 | 33 | | 2 (6.1%) | 7 (21.2%) | 15 (45.5%) | 9 (27.3%) | |
| 10.3.1_SouthAsia1 | 14 | | | **13 (92.9%)** | | 1 (7.1%) | |
| 10.3.2_MiddleEast1 | 16 | | 1 (6.3%) | 3 (18.8%) | | **12 (75.0%)** | |
| 10.3.3_Turkey2 | 10 | | | | | **10 (100%)** | |
| 10.3.4_Chile2 | 10 | | **10 (100%)** | | | | |
| 10.3.5_MiddleEast2 | 6 | | | | | **5 (83.3%)** | 1 (16.7%) |
| 10.3.6_SouthAmerica | 84 | | **79 (94.0%)** | | 3 (3.6%) | | 2 (2.4%) |
| 10.3.7_NorthAfrica2 | 51 | **41 (80.4%)** | | | 8 (15.7%) | | 2 (3.9%) |
| 10.3.8 | 10 | 1 (10.0%) | | | 4 (40.0%) | 5 (50.0%) | |
| 10.3.8.1_SouthAsia2 | 4 | | | **4 (100%)** | | | |
| 10.3.8.2_Turkey3 | 25 | | | | 1 (4.0%) | **24 (96.0%)** | |
| 10.3.8.3_MiddleEast3 | 5 | | | | | **5 (100%)** | |
| 10.3.8.4_MiddleEast4 | 8 | | | | | **8 (100%)** | |
| 10.3.8.5_EuropeNorthAfrica | 28 | 6 (21.4%) | | | **20 (71.4%)** | 1 (3.6%) | 1 (3.6%) |
| 10.3.9_NorthAfrica3 | 6 | **5 (83.3%)** | | | 1 (16.7%) | | |
| 11 | 1 | | | | **1 (100%)** | | |
| **Total** | **568** | 75 | 127 | 53 | 202 | 97 | 14 |

*n* number of isolates. *the isolates from Egypt were assigned to the Middle East. **the isolates from Turkey, Iraq, Lebanon, Syria, Mandatory Palestine, and Saudi Arabia were assigned to the Middle East. If a genotype is found at a percentage >50% in a particular geographic region, the data are indicated in bold.

found in three different HC50 clusters. The predominant cluster, HC50_1857, was not even specific to 10.3.6 (also found in isolates of five other genotypes) and the 10.3.6 isolates within HC50_1857, were further split between 15 different HC20 clusters, making them hard to track by cgMLST (Supplementary Figs 6 and 7, Supplementary Data 1).

### Evolution of antimicrobial resistance

The emergence of antimicrobial resistance (AMR) in SPB⁻ PG1 is recent (Fig. 5, Supplementary Data 1). Between 1898 and 2000, only one isolate (0.3%, 1/345) had antibiotic resistance genes (ARGs). This human isolate (B73-1117), collected in France in 1973, displayed resistance to ampicillin ($bla_{TEM-1D}$), streptomycin ($strAB$, $aadA1$, and $aadA2b$), sulfonamides ($sul1$), chloramphenicol ($cmlA1$), and tetracycline ($tetA$). Between 2001 and 2021, 23.1% (52/223) of isolates had

ARGs. One isolate acquired in Turkey in 2001 (01-7995) produced a CTX-M-3 extended-spectrum beta-lactamase[30], whereas another isolate (P7704) acquired in South America in 2019 produced an OXA-48 carbapenemase[31]. One isolate (83282), acquired in South America in 2014, contained the $mph(A)$ gene encoding a macrolide 2′ phosphotransferase, a common determinant of azithromycin resistance (however, the antimicrobial susceptibility pattern of this isolate was unavailable). All these rare isolates contained AMR plasmids (Supplementary Data 1); however, the mechanisms of resistance (21.5%, 48/223) most prevalent during the 2001–2021 period involved mutations of the quinolone resistance-determining regions of the chromosomal $gyrA$ and $gyrB$ DNA gyrase genes leading to resistance to nalidixic acid and/or decreased susceptibility or resistance to ciprofloxacin (minimum inhibitory concentration [MIC] between 0.125 and 0.5 mg/L). The

**Table 2 | Dating of the main lineages and genotypes of SPB⁻ PG1 with BEAST2**

| Main lineages and genotypes | Time (year) of the MRCA (95% HPD) |
|---|---|
| All 256 SPB⁻ PG1 | 1274 (915–1583) |
| Lineage 1 | 1757 (1592–1894) |
| Lineage 2 | 1813 (1745–1874) |
| Lineage 4 | 1740 (1641–1831) |
| Lineage 5 | 1727 (1660–1787) |
| Lineage 6 | 1853 (1797–1899) |
| Lineage 7 | 1765 (1709–1819) |
| 7.1.1_Chile1 | 1831 (1776–1887) |
| 7.2_EuropeEasternAsia | 1840 (1791–1882) |
| 7.3.1_NorthAfrica1 | 1896 (1861–1931) |
| 7.3.2_BAOR | 1896 (1866–1925) |
| Lineage 9 | 1768 (1711–1819) |
| 9.1_France | 1913 (1887–1933) |
| Lineage 10 | 1793 (1740–1842) |
| 10.3.1_SouthAsia1 | 1926 (1900–1950) |
| 10.3.2_MiddleEast1 | 1913 (1886–1937) |
| 10.3.4_Chile2 | 1946 (1921–1969) |
| 10.3.5_MiddleEast2 | 1930 (1904–1952) |
| 10.3.6_SouthAmerica | 1918 (1888–1942) |
| 10.3.7_NorthAfrica2 | 1932 (1916–1947) |
| 10.3.8.1_SouthAsia2 | 1993 (1978–2005) |
| 10.3.8.2_Turkey3 | 1994 (1985– 2002) |
| 10.3.9_NorthAfrica3 | 1930 (1907–1951) |

MRCA, most recent common ancestor; HPD, highest posterior density interval.

first isolate harbouring such a mutation was isolated in 2004. Diverse mutations were observed, with *gyrA*_S83F in 20 isolates (9.0%, 20/223), *gyrA*_D87Y in 12 isolates (5.4%, 12/223), *gyrA*_D87N in six isolates (2.7%, 6/223), *gyrA*_D82N in three isolates (1.3%, 3/223), *gyrB*_S464F in three isolates (1.3%, 3/223), *gyrA*_D87G in two isolates (0.9%, 2/223), a combination of *gyrA*_D87N and *gyrB*_S464F in one isolate (0.4%, 1/223), and a combination of *gyrA*_S83F and *gyrA*_D87G in one isolate (0.4%, 1/223). For this last isolate (CLA-76-5), the antimicrobial susceptibility pattern of which was unavailable, we cannot rule out resistance to ciprofloxacin (MIC > 0.5 mg/L). These *gyrA* and *gyrB* mutations occurred over the entire phylogenetic tree, in lineages L2 ($n = 4$), L3 ($n = 1$), L5 ($n = 2$), L7 ($n = 3$), L9 ($n = 5$), and L10 ($n = 33$) (Fig. 5). The lineage L10 isolates containing such *gyrA* and *gyrB* mutations were further classified into eight different genotypes. One cluster contained seven *gyrA*_D87Y isolates acquired in Turkey between 2009 and 2017 (genotype 10.3.8.2_Turkey3) (Supplementary Data 1).

**Pan-genome analysis and *sopE* prophages**
A pan-genome analysis of the 568 SPB⁻ PG1 genomes studied (Supplementary Fig. 8) identified a closed pan-genome (alpha = 1.3) containing a total of 5,681 genes, including a core genome of 4044 genes (Supplementary Data 3).

Of the 1506 accessory genes present in <95% of the genomes, 696 (46.2%), 242 (16.1%), and 28 (1.9%) were found to belong to prophages, plasmids (with or without AMR genes), and transposases, respectively (Supplementary Data 4). The mapping of these accessory genes onto the core genome phylogeny revealed an absence of phylogenetic clustering for plasmid genes, whereas some prophage genes were (sub)lineage-specific (Supplementary Fig. 9). We then evaluated the occupancy of the 10 prophage insertion sites – identified in the 14 complete genomes – with short-read assemblies from the entire diversity dataset (Supplementary Note "Prophages of SPB⁻ PG1", Supplementary Fig. 10).

Three (sites #1 to 3) of the 10 sites were occupied by prophages containing *sopE*, a virulence gene that encodes an effector translocated into eukaryotic cells that promotes bacterial invasion through cytoskeleton rearrangement. The nucleotide sequence of the *sopE* gene was 100% identical in the 14 complete genomes, regardless of the prophage into which it was inserted, with the exception of the *sopE* gene inserted at site #3 of the B624 genome, which differed from the *sopE* consensus sequence of SPB⁻ by eight (five being non-synonymous) of the 723 nucleotides. Two types of *sopE* prophages, both belonging to class *Caudoviricetes*, were identified; the first was 41,250 bp to 44,615 bp in size and displayed 89.8% to 95.9% nucleotide identity (38% to 55% coverage) to the *Salmonella Brunovirus* SEN34 (GenBank accession no. NC_028699); the second was 34,723 bp long and displayed ~98% nucleotide identity (83% coverage) to the enterobacterial *Xuanwuvirus* P88 (GenBank accession no. NC_026014) (Supplementary Table 1).

The *sopE*-containing SEN34-like prophages occupied sites #1 and #2, whereas the P88-like prophage occupied site #3 (Fig. 6a). This *sopE*-prophage content was correlated with SPB⁻ phylogeny, with a *sopE*-containing SEN34-like prophage inserted at site #1 in lineages L1 to L4 and at site #2 in lineages L5 to L11 (Fig. 6b). The P88-like prophage was not seen alone but always in addition to the SEN34-like prophage and its presence was strongly associated with the Dundee PT (Chi squared test, $p < 0.0001$) (Supplementary Fig. 10). All but two (173-73 and 73-67) of the 568 genomes had at least one *sopE* prophage: most genomes (75.1%, 425/566) had one, some (17.8%, 101/566) had two and a few (7.1%, 40/566) had three (Fig. 6). All genomes containing three *sopE* prophages belonged to genotype 9.1_France (associated with the Dundee PT).

**Development of a new SNV-based genotyping tool and analysis of recent SPB⁻ isolates**
We identified marker SNVs unique to each genotype (38 SNVs in total) (Supplementary Data 5) and implemented this genotyping scheme on the open-source Mykrobe platform (https://www.mykrobe.com/). Mykrobe checks (i) for the presence of *invA* to ensure that the genome belongs to the genus *Salmonella*, (ii) and then for the presence of the STM 3356 *d*-Tar⁻ specific SNV to confirm that the genome belongs to SPB⁻, and (iii) finally assigns genomes to genotypes based on presence of the 38 genotype-specific SNVs. After validation of the Mykrobe-implemented version of the scheme on the 568 genomes from the diversity dataset (Supplementary Note "Development of a new SNV-based genotyping tool for SPB⁻ PG1"), this genotyping tool was used on the 336 genomes from the surveillance dataset – originating from public health laboratories in the UK ($n = 200$), France ($n = 84$), USA ($n = 39$), and Canada ($n = 13$), with isolation dates between 2014 and 2023 – to identify genotypes recently isolated in these four countries (Fig. 2c, Supplementary Data 6). During the 2014 – 2023 period, 25 genotypes were observed. Genotype diversity was highest among French isolates (17 genotypes for 84 isolates) and lowest in the UK (13 genotypes for 200 isolates). The most frequent genotype was 10.3.6_SouthAmerica, found in 49.1% (164/336) of the isolates from all four countries. Its frequency ranged from 44% (125/284) for the isolates collected in Europe to 76.9% (40/52) for the isolates collected in North America. The second most frequent genotype, 10.3.2_MiddleEast1 (18.8%, 63/336), was found mostly in the UK. The other genotypes were found in less than 10% of the isolates. There was an overrepresentation of genotypes linked to South Asia in the UK (e.g., 10.3.1_SouthAsia1 and 10.3.8.1_SouthAsia2, together accounting for 11% (22/200) of the isolates), and of genotypes linked to North Africa in France (e.g., 7.3.1_NorthAfrica1, 7.3.2_BAOR, 10.3.7_NorthAfrica2 and 10.3.9_NorthAfrica3, together accounting for 13.1% (11/84) of the isolates). Interestingly, older genotypes – such as 2.1, accounting for 3.5% (7/200) of UK isolates, and 9.1_France, accounting for 9.5% (8/84) of French isolates – are still being isolated.

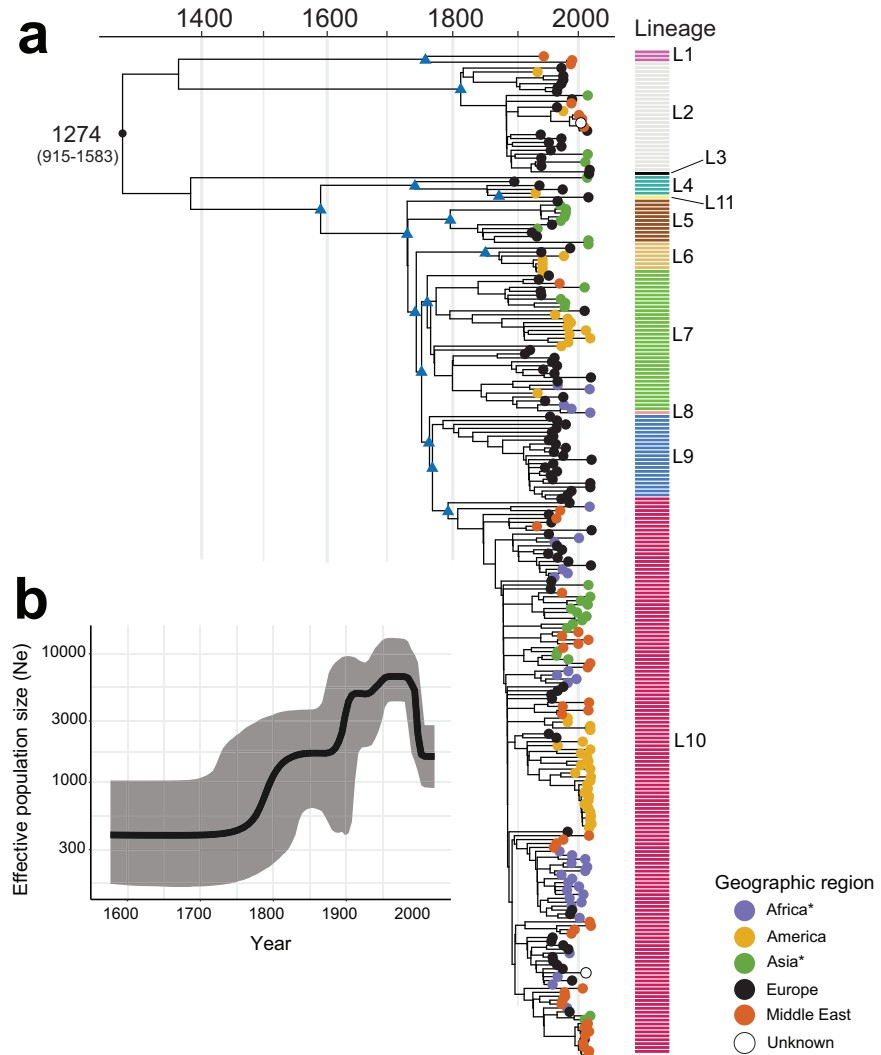

**Fig. 3 | Timed phylogeny of a representative subsample of 256 SPB⁻ PG1 isolates.**
**a** Maximum clade credibility tree produced with BEAST2 (optimised relaxed clock model; Bayesian skyline) with the tips coloured according to the geographic origin of the isolates (see inset). Selected nodes supported by posterior probability values > 0.9 are shown as blue triangles. The estimated age of the MRCA (with 95% confidence intervals in parentheses) is shown. The lineage of these isolates is indicated at the right side of the tree. The scale bar indicates the number of substitutions per variable site (SNVs). An asterisk indicates the reassignment of some African and Asian countries to the Middle East (see Table 2). **b** Bayesian skyline plot showing temporal changes in effective population size (Ne) (black curve) with 95% confidence intervals (grey shading).

## Discussion

Global phylogenomic studies of bacterial pathogens can be strongly affected by sampling biases, such as a lack of bacterial strains from certain geographic areas and periods of time. We tried to minimise these sampling biases, by ensuring that we studied a spatiotemporally representative set of SPB⁻ strains selected after (i) a comprehensive search of the medical and scientific literature since the first report of PTB, (ii) an analysis of SPB⁻ strains available from an international network of reference laboratories with collections of historical isolates, and (iii) a search for SPB⁻ genomes among the >400,000 *Salmonella* genomes present in the EnteroBase database.

Based on our dataset of 568 SPB⁻ PG1 genomes, we estimated the age of this pathogen at ~750 years (1274 CE; 95% CI, 915–1583), which is very close to the previous median date of origin estimated by Connor and coworkers[27] (1188 CE; 95% CI, 469 BC – 1799 CE), who used only 25 SPB⁻ PG1 genomes (i.e., those with a known year of isolation). SPB⁻ is older than SPA, which is estimated to have originated 450–700 years ago[25]. SPA was discovered two years after SPB (in the USA in 1898)[1,2] but is currently the most frequent agent of paratyphoid fever[28]. Due to lineage extinction, in particular, times to the MRCA are often

underestimated and the inclusion of ancient DNA in the analysis would increase precision and make it possible to establish dates of origin further in the past[25]. The dating of a representative collection of modern isolates of SPC estimated the origin of SPC to 456–664 years ago. When a draft SPC genome from an 800-year-old Norwegian skeleton was added to the analysis, the time to the MRCA increased to 1162–1526 years[26]. Unfortunately, no ancient DNA is currently available for SPB⁻ strains.

SPB⁻ strains, first identified in Europe in 1896, became a common cause of enteric fever across Europe thereafter, suggesting that they were circulating in this particular region of the world. Indeed, SPB⁻ genetic diversity was greatest among European isolates and those collected in Turkey. Turkey, part of which lies in Europe, has a long shared history with the rest of Europe through the rule of the Ottoman Empire over most of the countries in south-eastern Europe for several centuries. SPB⁻ isolates were, however, less frequently reported in other parts of the world (the Americas, East Asia, North Africa) early in the 20th century. In Morocco, North Africa, SPB⁻ strains were reported to have been introduced around the city of Fes by the French troops during the Rif war in 1925 (ref. 32). Two genotypes may have been

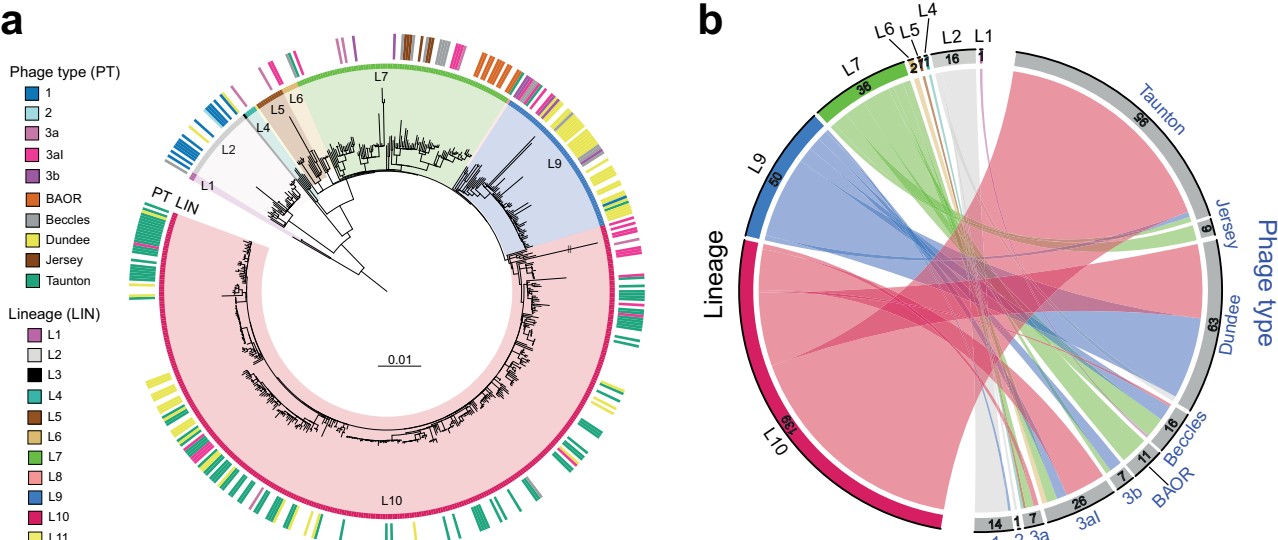

**Fig. 4 | Correlation between genome and phage-typing data for SPB⁻ PG1.**
**a** Maximum likelihood phylogeny (as in Fig. 1a) showing the 254 isolates from the diversity dataset that were phage-typed and their phage types (PT, see legend, inset). **b** Circular plot illustrating the correspondence between phage type and lineage for each of these 254 isolates. The flow chart bars are coloured according to the lineage (see the legend in the inset of panel "a"). The number of isolates is also indicated for each phage type and lineage. Source data are provided as a Source Data file.

introduced into Morocco at this time: 7.3.1_NorthAfrica1 or 7.3.2_BAOR. Other introductions from Europe may have occurred for older, pre-L10 lineages, including the introduction of genotypes 2, 5 and 7.2_EuropeEasternAsia in East Asia, and of genotype 7.1.1_Chile1 in Chile.

Our data for historical isolates and a careful exploration of the world surveys of PTs enabled us to determine which genotypes were present during the first half of the 20th century. In the UK, the genotypes associated with domestic infections[16] were probably 2.0 and 2.1 (representing PTs 1 and 2) and 6 and 7.2_EuropeEasternAsia (representing PT 3). However, many rare PTs were introduced into the UK after WWII following the return of millions of demobilized soldiers and the reopening of tourist traffic[16]. Taunton predominated among these new PTs, and included various genotypes of lineage L10, mostly found in non-European isolates (from North Africa, South Asia, the Middle East and South America). In France, one epidemic strain of PT Jersey emerged in Western France in 1951 (only 0.13%, 1/752 of the isolates were phage-typed as Jersey in 1950), peaked in 1952 (60.1%, 252/419 of the isolates in this year), and then became very rare in the 1970s[12,18,33]. This strain could be assigned to genotype 7.3 (monophasic SPB⁻, see Supplementary Note "Prophages of SPB⁻ PG1"). Widespread outbreaks caused by a strain phage-typed as Dundee also occurred in France during the spring and summer of 1949 (ref. 16). This strain, assigned to genotype 9.1_France, remained prevalent in France until the mid-1980s[10–12,16–18,33].

In Europe, improvements in hygiene following WWII, including food hygiene (control measures for foods at risk implemented in bakeries in the UK), sanitation, and access to safe water and antibiotics, probably prevented the spread of SPB⁻. Ultimately, this led to a steady decrease in the prevalence of local PTB, with these infections increasingly confined to travellers or migrants returning from the Middle East, North Africa, South America, and South Asia. Our analysis of recent routine SPB⁻ PG1 isolates sequenced over the last 10 years in France, the UK, the US, and Canada indicated that the most frequent genotype (~50% of all these isolates) is currently 10.3.6_SouthAmerica. This genotype is mostly associated with Andean countries from western South America such as Peru, Bolivia and, more recently, Argentina. We estimated that this genotype was introduced into South America,

probably from Europe, between 1899 (95%CI: 1871–1927) and 1918 (95% CI: 1888–1942). Between 1973 and 1977, only 1% (2/200) of the imported PTB cases in the UK were acquired in South America[14], and in 2023, genotype 10.3.6_SouthAmerica accounted for 78.1% (25/32) of all PTB cases detected in France. In Argentina, an increasing number of PTB cases (total of ~5500 cases) have been reported in the Salta province (bordering Bolivia) since 2018 (ref. 34), but no such epidemiological pattern was found in Peru and Bolivia. It is therefore important to conduct local studies aiming to identify the areas of current PTB transmission (probably in remote regions with tourist sites) and associated risk factors, to facilitate mitigation. Interestingly, old genotypes are still being isolated in Europe. For example, 4% of the surveillance isolates from the UK belonged to genotypes 2.1 (*n* = 7) and 5 (*n* = 1), and 9.5% (*n* = 8) of those in France belonged to the 9.1_France genotype. The eight French cases, for which isolates were not recovered from blood samples, were patients between 81 and 98 years of age with no recent history of travel to countries in which PTB is endemic, suggesting that they may be long-term carriers infected several decades ago. We can therefore conclude that the distribution of SPB⁻ PG1 genotypes within these four high-income countries in recent years reflects (i) the destinations of their holidaymakers, (ii) movements of people linked to migration and travel patterns over recent centuries, and finally (iii) remnants of past infections in long-term carriers.

One feature particular to SPB⁻ strains is their sensitivity to antimicrobial drugs, which is greater than that of other agents of enteric fevers, such as STY[24] and, to a lesser extent, SPA strains[25]. AMR mostly concerned quinolones and their resistance determinants, emerging over the last 20 years, without horizontal transmission. Different mutations of the chromosomal *gyrA* gene occurred in many genotypes across the entire phylogeny, in isolates from Europe, South America, East and South Asia, and the Middle East, suggesting a high level of fluoroquinolone exposure worldwide. Long-term SPB⁻ PG1 carriers may have particularly high levels of exposure; for example, three of the eight (37.5%) genotype 9.1 isolates recently obtained from elderly French patients (see above) had *gyrA* mutations (of three different types). However, no successful AMR SPB⁻ strains have emerged, contrasting with the situation for STY, the H58 clone of which emerged in South Asia in the 1980s before spreading globally[35,36]. SPB⁻ strains are

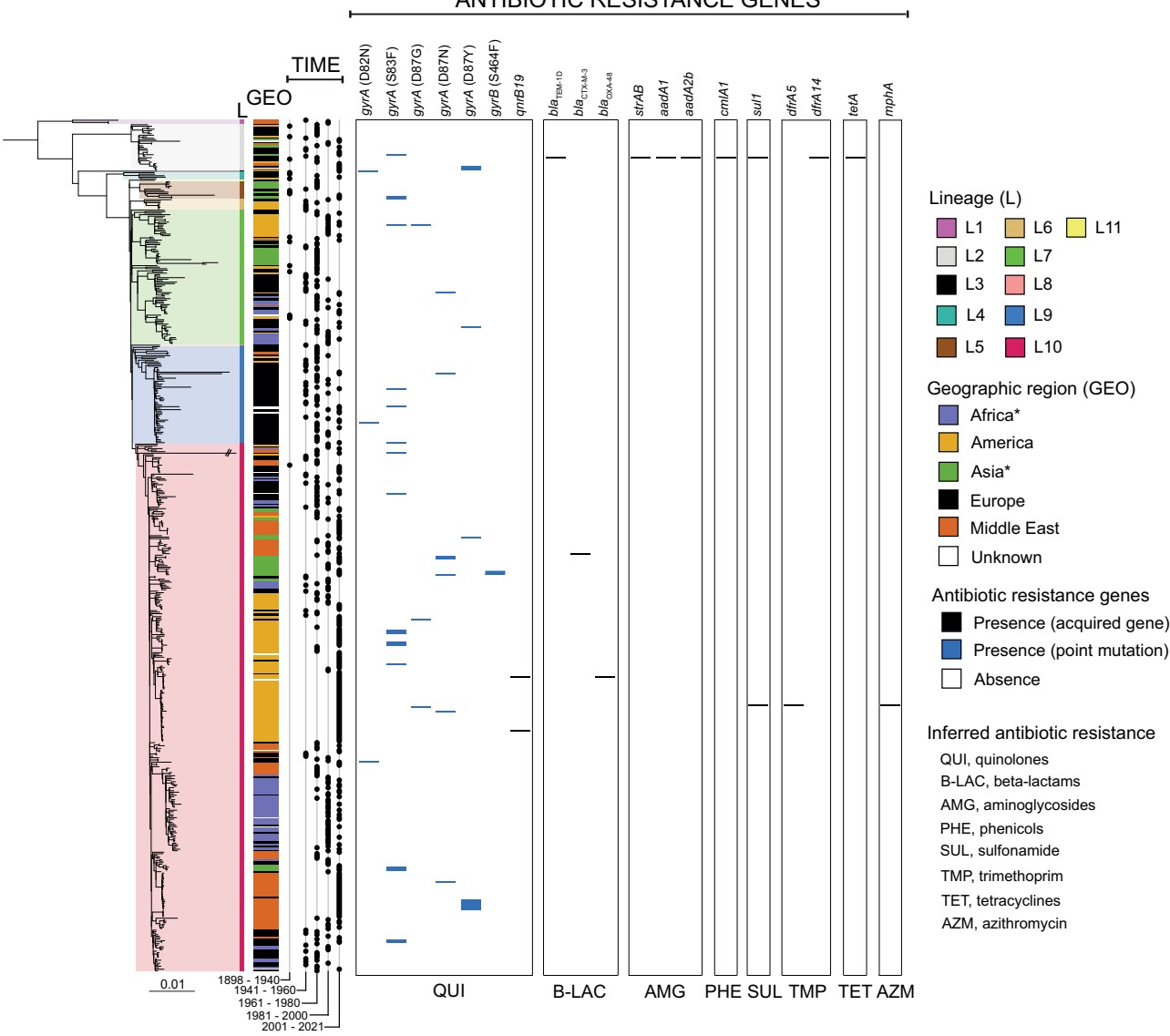

**Fig. 5 | Genomic characterization of antibiotic resistance genes in SPB⁻ PG1.**
Distribution of antibiotic resistance genes by phylogeny (as in Fig. 3a), geography, and time period. Genes acquired via horizontal gene transfer are indicated in black and those acquired via chromosomal mutation are indicated in blue. An asterisk indicates the reassignment of some African and Asian countries to the Middle East (Table 1).

not very frequent and have only recently been detected in South Asia — where selection pressures exerted by quinolones and, later, fluoroquinolones, began early — potentially accounting for this lower level of AMR than in STY and SPA strains.

The presence of a particular *sopE* bacteriophage (ΦSopE309) was previously used by Prager and coworkers[37] to distinguish systemic SPB⁻ isolates from enteric SPB⁺ isolates. By contrast, Connor and coworkers[27] suggested that the *sopE* gene was not a suitable marker for identifying SPB⁻ PG1 isolates because this gene was not present in all of their PG1 isolates and some of these isolates had a gene homologous to *sopE* also found in all other SPB PGs. As ΦSopE309 was initially isolated from SPB⁻ strain B309 — also included in our study — we were able to determine that it was actually the Brunovirus SEN34-like prophage. Our in-depth analysis supports the recommendation of Prager and coworkers[37] to use *sopE* as a marker gene for SPB⁻ because the *sopE*-carrying SEN34-like prophage was present in almost all the 568 SPB⁻ PG1 isolates from our diversity dataset (including 28 genomes from the study by Connor and coworkers[27]). Only two isolates were devoid of

*sopE* and each of the three insertion sites for *sopE* prophages were empty in these isolates. These two isolates were collected more than 50 years ago and we cannot therefore rule out the possibility that the *sopE* prophage was excised during storage or subculture. Furthermore, according to their nearest neighbours on the phylogenetic tree, these isolates would have contained only one *sopE* prophage. The gene homologous to *sopE* unexpectedly found in some SPB⁻ PG1 genomes by Connor and coworkers[27] is almost certainly *sopE2*, a chromosomal gene present in all S*almonella* lineages and encoding an effector protein, SopE2, 69% identical to SopE[38]. By contrast, the *sopE* gene, carried by different bacteriophages (lambda-like Gifsy-2, P2-like, mTmV), was previously found in only some serotypes of *Salmonella* (e.g., Gallinarum, Typhi, Dublin, Heidelberg) or some strains of certain serotypes (e.g., epidemic *S. enterica* serotype Typhimurium DT204 in the 1970s or the currently dominant monophasic *S. enterica* serotype Typhimurium ST34)[39–41]. It has been suggested that the acquisition of *sopE* by lysogenic conversion increases the fitness of *S. enterica* serotype Typhimurium, thereby contributing to the emergence of epidemic

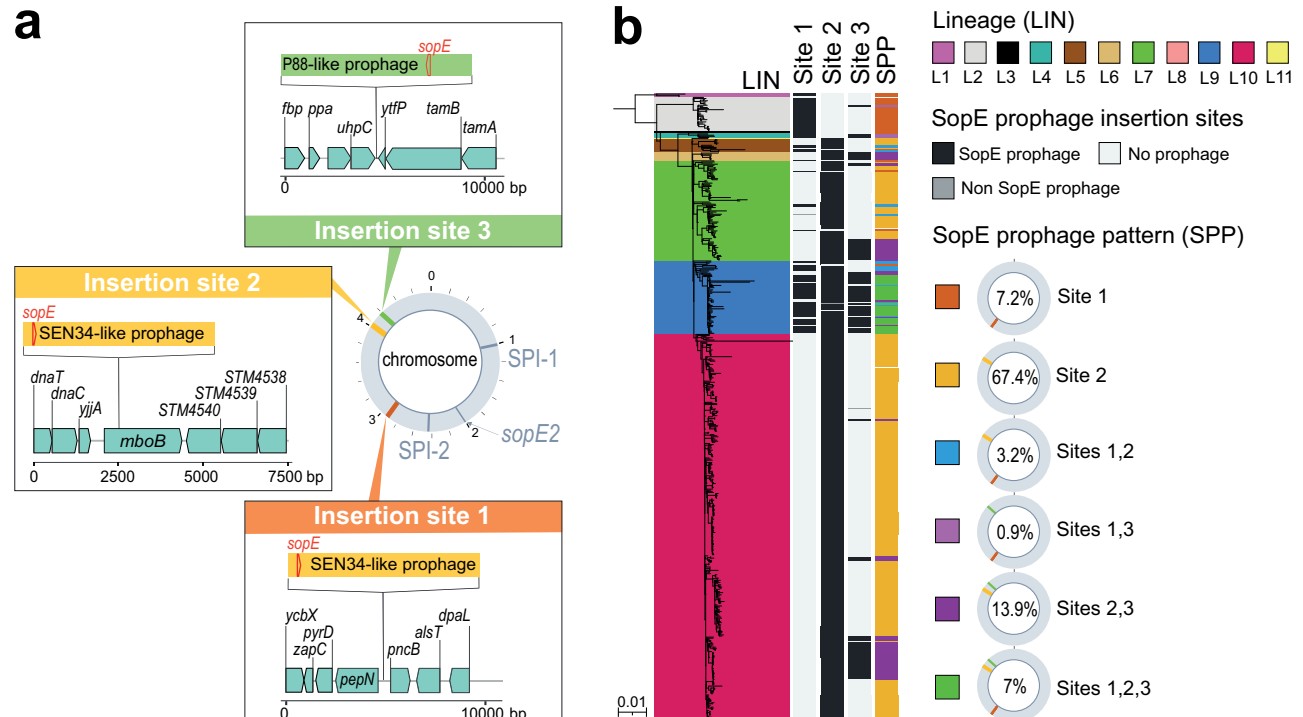

**Fig. 6 | Lineage-specific accumulation of *sopE* prophages in SPB⁻ PG1. a** Three insertion sites occupied by *sopE* prophages were identified in the 14 complete genomes of SPB⁻ PG1 isolates. Prophages of the SEN34 family [40.89–44.3 kb] were found at insertion sites #1 and/or #2. The P88-family prophage [34.3 kb] was found at insertion site #3. Further details concerning the insertion sites are available from Supplementary Data 8. Gene arrow maps were generated with the gggenes v.0.5.0 and ggplot2 v.3.4.2 packages of R v.4.1.2 software. **b** The 568 genomes from the diversity dataset were screened for an absence of insertions at sites #1, #2 and #3. Presence/absence is colour-coded in black and light grey, respectively; dark grey indicates the presence of a potential *sopE*-free prophage. Six types of prophage insertion were recorded across the 11 lineages (Supplementary Data 9).

strains[42]. SopE and SopE2 are G-nucleotide exchange factors that are translocated into the host cell, where they activate host cellular Rho-GTPases – such as Cdc42 and Rac1 for SopE and Cdc42 alone for SopE2 – which act as key regulators of diverse activities, such as the expression of pro-inflammatory cytokines and the organisation of the actin cytoskeleton, ultimately promoting bacterial uptake and even intracellular replication[42,43]. In addition to their chromosomal *sopE2* gene and their SEN-34-like prophage *sopE* gene, 17.8% and 7.1% of SPB⁻ PG1 isolates contain one and two additional copies of *sopE* (carried by either the SEN-34-like or P88-like prophages), respectively. Three copies of *sopE* were observed exclusively in the PT Dundee strain (genotype 9.1_France) that was epidemic in France after WWII and is still isolated, even now, from long-term carriers. Similarly, two copies of *sopE* were previously described in *S. enterica* serotype Heidelberg isolates from a large multistate outbreak linked to turkey meat in the US[44] and three copies were found in a cluster of monophasic *S. enterica* serotype Typhimurium ST34 isolates from humans and pigs in the UK[41]. An increase in *sopE* copy number is observed in ~25% of SPB⁻ PG1 isolates, but additional studies are required to determine the consequences of this increase for the expression of *sopE* and interaction with host cells.

We also provide a new robust framework for identifying and tracking the causal agent of PTB. Firstly, SPB isolates can be assigned to the 10 known PGs described by Connor and coworkers[27] with the EnteroBase cgMLST scheme, with the HC400_1620 cluster considered a signature of SPB⁻ PG1. This cluster has been shown to be more informative than the traditional MLST7 ST86 criterion. Even though ST86 is an excellent predictor of SPB⁻ PG1 isolates, the existence in such isolates of other SLVs of ST86 (such as ST772, ST2340, ST5113, ST6558, ST7999, ST8505, ST8506, ST8931, ST8932, ST9224, ST10005, ST10013, and ST10014 and potentially others), a triple-locus variant of

ST86 (ST2134), or even the possibility of non-typability by MLST might complicate the use of MLST7 as a unique tool for identifying SPB⁻ PG1. Furthermore, the SLVs of ST86 are also observed in other SPB PGs, such as ST43 in PG3 and PG4, and ST149 in PG5 (ref. 27). If MLST7 is used, we recommend its use in combination with detection of the specific *d*-Tar⁻ SNV within STM 3356, this SNV being crucial for the formal assignment of non-ST86 genomes to SPB⁻ PG1. We have also developed a hierarchical SNV-based genotyping scheme for tracking the different SPB⁻ PG1 strains, an approach previously successfully used for the surveillance of the two main agents of enteric fever, STY[45] and SPA[46]. As already reported for *S. sonnei*[47], we showed that our SNV-based genotyping scheme performs better than cgMLST hierarchical clustering at higher levels of resolution for tracking epidemiologically relevant strains. Our scheme, now implemented in the open-source Mykrobe software (https://github.com/mykrobe-tools/mykrobe), can identify 38 different genotypes with a phylogenetically – and sometimes phylogeographically – informative nomenclature. It is, however, important to apply this scheme only to confirmed SPB⁻ PG1 genomes – identified by the cgMLST HC400_1620 cluster or using MLST7 plus the specific *d*-Tar⁻ SNV – as Mykrobe may otherwise assign the rare genotype 7.0 and to a lesser extent genotype 7.3 to most non-PG1 *Salmonella* genomes (for non-PG1 genomes the tool will, however, always yield "Unknown" instead of "Salmonella_Paratyphi B" under the column "species" of the output table). We anticipate that the use of this scheme and its universal nomenclature will improve the laboratory surveillance of PTB. It is now easier to track the different SPB⁻ PG1 populations at global scale. We were able to identify the different genotypes detected recently in travellers from or migrants to four high-income countries in North America and Europe (surveillance dataset). However, more global studies involving countries experiencing PTB across the globe and performed regularly over time – as for

the PT surveys − would be helpful to monitor the diversity, spread and evolution (of AMR in particular) of this pathogen. New genotypes with geographic information (in cases of new emerging genotypes in a defined area) or new updates on alias names (in cases of the establishment of known genotypes in new areas) should be added to the scheme in the future. The use of a common nomenclature would also be helpful during outbreak investigations. It is now straightforward to define the bacterial types of SPB⁻ PG1 and to share this information during transborder outbreak investigations. This should also facilitate the identification of transmission chains associated with particular geographic regions, particularly in areas in which PTB is not endemic. For example, genomic surveillance in the UK identified an imported SPB⁻ outbreak in travellers coinciding with a mass gathering in Iraq in 2021 (ref. 29). The isolates from these patients clustered in one of the two clades labelled "travel to Iraq". According to our genotyping scheme, this clade corresponds to genotype 10.3.2_MiddleEast1. The oldest isolates belonging to this genotype were collected in Iran in 1965 and Iraq in 1975, suggesting that this strain has been endemic in the region for many decades. The second clade labelled "travel to Iraq" identified by UKHSA corresponded to genotype 10.3.8.3_MiddleEast3. In endemic regions, this genotyping scheme might also be useful for determining the genotype of a potential outbreak strain if several genotypes are known to circulate regionally.

In conclusion, using a carefully selected set of genomes from historical and contemporary isolates, we were able to unravel the population structure and evolution of SPB⁻ PG1, the agent of PTB. This pathogen, which emerged at least 750 years ago, initially thrived in Europe, but is now active in other parts of the world, frequently in areas lacking enteric fever surveillance systems. We anticipate that the widespread use of our genotyping scheme by public health laboratories will improve our understanding of the global epidemiology of this pathogen.

## Methods

### Ethics statement

This study was based exclusively on bacterial isolates (including historical and reference strains) and associated metadata. The vast majority of these isolates ($n = 362$) were obtained from *Institut Pasteur* (reference laboratories or *Collection de l'Institut Pasteur* (CIP)); 115 isolates corresponded to historical or reference strains collected between 1898 and 1971 and 247 were bacterial isolates collected under the mandate for laboratory-based surveillance awarded by the French Ministry of Health to the National Reference Centre for *Escherichia coli*, *Shigella* and *Salmonella* (NRC-ESS) since 1972. Data collection and storage by the NRC-ESS was approved by the French National Commission for Data Protection and Liberties ("*Commission Nationale Informatique et Libertés* (CNIL)"; approval number: 1474659). For other human bacterial isolates collected (or the genome sequences derived from them) by participating reference laboratories under local mandates for laboratory-based surveillance of salmonellosis in line with local laws and regulations, the associated metadata did not contain any personal identifiable information and were restricted to year and country of isolation, and international travel information. As a result, neither informed consent nor approval from an ethics committee was sought given this is not research conducted on human participants.

### *S. enterica* serotype paratyphi B sequence data collections

**Diversity dataset.** We first studied a diversity dataset of 568 SPB⁻ genomes, 446 of which were generated specifically for this study, 109 had already been published[27−29,31,48−50], and the other 13 were unpublished but deposited in EnteroBase (https://enterobase.warwick.ac.uk/species/index/senterica) or GenBank (https://www.ncbi.nlm.nih.gov/genbank/) (Supplementary Data 1). This diversity dataset also contained the reference strain (116 K) of *S. enterica* serotype Onarimon

(1,9,12:b:1,2) considered an O-antigen variant of SPB⁻ (Supplementary Note "Validation of the SPB⁻ PG1 diversity dataset").

The 446 isolates were obtained from the bacterial collections of *Salmonella* reference laboratories located at the *Institut Pasteur* (IP), Paris, France ($n = 351$), NHS Greater Glasgow and Clyde (NHSGGC), Glasgow, UK ($n = 19$), the National Institute of Infectious Diseases (NIID), Tokyo, Japan ($n = 18$), the Centers for Disease Control and Prevention (CDC), Atlanta, GA, USA ($n = 15$), the Pasteur Institute of St Petersburg (PISP), St Petersburg, Russian Federation ($n = 12$), the UK Health Security Agency (UKHSA), Colindale, UK ($n = 10$), the Robert Koch Institute (RKI), Wernigerode, Germany ($n = 7$), University College Hospital (UCH), Galway, Ireland ($n = 3$), or from the *Collection de l'Institut Pasteur* (CIP) ($n = 11$).

The 568 genomes from the diversity dataset were obtained from isolates collected between 1898 and 2021 from humans (472/568, 83.1%), environmental samples (42/568, 7.4%), animals (14/568, 2.5%), food items (2/568, 0.3%) or unknown sources (38/568, 6.7%) (Supplementary Data 1). For the human isolates, 43% (203/472) were obtained from blood, 39.2% (185/472) from stools, 3.8% (18/472) from other sources (urine, cerebrospinal fluid, pus, cysts, wounds, gallbladders, bile) and 14% (66/472) were of unknown origin. The 568 isolates and strains were isolated locally or from travellers and originated from 41 countries in Europe (202/568, 35.6%), Asia (144/568, 25.3%), the Americas (127/568, 22.3%), and Africa (81/568, 13.2%). Some historical or laboratory strains were of unknown geographic origin (14/568, 2.5%). The European isolates accounted for 57.9% (135/233) of the total isolates collected in the 1898−1980 period, decreasing to 20% (67/335) in the 1981 − 2022 period.

**Surveillance dataset.** We assembled a surveillance dataset of 336 SPB⁻ genomes routinely obtained by four public health microbiology laboratories (CDC, UKHSA, IP and Canada's National Microbiology Laboratory) and submitted to EnteroBase between August 27th 2015 and May 2nd 2023. These 336 genomes included 111 already present in the "diversity dataset" (Supplementary Data 6).

### Antimicrobial drug susceptibility testing

Antimicrobial drug susceptibility was determined at IP for 273 SPB⁻ isolates from the diversity dataset (Supplementary Data 1) by disk diffusion on Mueller-Hinton (MH) agar in accordance with the guidelines of the Antibiogram Committee of the French Society for Microbiology (CA-SFM) / European Committee on Antimicrobial Susceptibility Testing (EUCAST)[51]. The following antimicrobial drugs (Bio-Rad, Marnes-la-Coquette, France) were tested: amoxicillin, ceftriaxone, ceftazidime, streptomycin, kanamycin, amikacin, gentamicin, nalidixic acid, ofloxacin, ciprofloxacin, sulfonamides, trimethoprim, sulfamethoxazole-trimethoprim, chloramphenicol, tetracycline, and azithromycin. *Escherichia coli* CIP 76.24 (ATCC 25922) was used as a control. The minimal inhibitory concentrations (MICs) of nalidixic acid and ciprofloxacin were determined for 84 isolates (all 24 isolates resistant to nalidixic acid in disk diffusion tests and 60 susceptible isolates chosen on the basis of year of isolation and antibiotic resistance gene content) with Etest strips (bioMérieux, Marcy L'Etoile, France). As a means of distinguishing *Salmonella* isolates susceptible to ciprofloxacin (minimum inhibitory concentration [MIC] ≤ 0.25 mg/L), which are wild-type (WT), from those that are non-WT, we defined two categories based on the epidemiological cutoffs used by the CLSI: decreased susceptibility to ciprofloxacin (MIC between 0.12 and 0.5 mg/L) and true susceptibility to ciprofloxacin (MIC ≤ 0.06 mg/L)[52]. Please note that due to clinical evidence of poor response to ciprofloxacin in systemic infections caused by *Salmonella* spp. isolates displaying such decreased susceptibility to ciprofloxacin, the clinical breakpoint to define ciprofloxacin resistance in non-enteric isolates of this species is MIC > 0.06 mg/L[51].

## Short-read sequencing

Illumina short-read sequencing was performed by the IP genomics platforms (PF1 and the Mutualised Platform for Microbiology (P2M)) for 389 isolates (362 from IP plus 12 from PISP and 15 from the CDC). At PF1 (150 isolates sequenced), total DNA was extracted with the Wizard Genomic DNA Kit (Promega, Madison, WI, USA) and fragmented with a Covaris E220 ultrasonicator. Sequencing libraries were then prepared with the NEXTflex PCR-free DNA-Seq Kit (Bioo Scientific Corporation, Austin, TX, USA) and sequencing was performed with the HiSeq 2500 system (Illumina, San Diego, CA, USA). At P2M (239 isolates sequenced), total DNA was extracted with the MagNA Pure DNA isolation kit (Roche Molecular Systems, Indianapolis, IN, USA). Sequencing libraries were prepared with the Nextera XT kit (Illumina), and sequencing was performed with the NextSeq 500 system (Illumina). The other 57 SPB⁻ isolates were sequenced by the participating laboratories, in accordance with their usual practices (Supplementary Methods "Short-read sequencing").

Paired-end reads varied in length according to the sequencing platform/site, from 95 to 300 bp, yielding a mean coverage of 124-fold for each isolate (minimum 25-fold, maximum 350-fold) (Supplementary Data 1).

Taxonomic read classification with Kraken[53] v.2.1.1 was used to confirm that sequencing reads originated from *Salmonella* and not from a contaminant. All short-read data are publicly available. Their genome accession numbers are provided in Supplementary Data 1.

## Long-read sequencing and complete genome circularisation

At the time of the study, there was no complete SPB⁻ PG1 genome that could be used as a reference genome. We therefore sent total DNA extracted from strain CIP 54.115 (ref. 27) to GATC Biotech (now Eurofins Genomics) for long-read sequencing on the Pacific BioSciences RSII platform. The high-quality de novo assembly was performed by GATC Biotech after the hierarchical genome-assembly process (HGAP)[54]. We performed an additional step of assembly polishing with short reads and the Pilon[55] v.1.23 tool.

Long-read sequences for another 11 isolates − selected to provide a maximal representation of diversity in terms of lineages and phage types − were generated with Oxford Nanopore Technology (ONT) (Supplementary Data 1). Bacterial DNA was extracted from cultures grown in Trypto-casein soy broth (Bio-Rad) at 37 °C with shaking at 200 rpm. We used Genomic-tip 100/G columns (Qiagen) according to the manufacturer's instructions, for DNA extraction. DNA integrity and the absence of RNA were checked by agarose gel electrophoresis and by the determination of $A_{260}/A_{230}$ and $A_{260}/A_{280}$ ratios with a NanoDrop™ 2000 spectrophotometer. DNA concentrations were measured with the Qubit system and the dsDNA BR Assay Kit (Invitrogen). Libraries were prepared from total DNA with the SQK-LSK109 ligation sequencing kit and the EXP-NBD104/114 barcoding kit according to the ONT procedure (Native Barcoding Amplicons protocol version ACDE_9064_v109_revP_14Aug2019, https://doi.org/10.17504/protocols.io.bgzxjx7n). Libraries were sequenced with R9.4.1 flow cells and a Mk1C MinION sequencer. Base calling was performed with Guppy[56] v.4.3.4 or v.5.0.13. The filtlong tool (https://github.com/rrwick/Filtlong/) v.0.2.1 was used to filter reads according to their length (min_length 800 bp) and quality (keep_percent 90). Read lengths ranged from 3827 to 20,347 bp (mean of 9,923 bp), with a mean of 262-fold coverage per genome (minimum 54-fold, maximum 676-fold). Complete de novo genome assemblies were generated with the Trycycler[57] v.0.5.0 pipeline using default parameters. For each isolate, long reads were subsampled into 12 sets, which were subsequently used to generate 12 independent assemblies (four sets per assembler) with Flye[58] v.2.9, raven[59] v.1.6.0 or miniasm[60] v.0.3-&-Minipolish[60] v.0.1.3. The consensus assembly was first long read-polished with medaka v.1.4.4 (https://github.com/nanoporetech/medaka) and then short read-polished four times with pilon[55] v.1.23. The final assemblies were annotated with bakta[61] v.1.5.0.

## Other genomes

SPB⁺ PG2 strain 201906085 (ENA accession no. ERR12749341) and SPB⁺ PG3 strain SPB7 (GenBank accession no. NC_010102.1) were used as outgroups to identify the ancestral lineage of SPB⁻ PG1 genomes (Supplementary Data 1).

## Genomic typing methods

A two-step in silico serotype confirmation procedure was used: O-antigen determination by a fast kmer-alignment method from KMA[62] v.1.4.14 for the alignment of raw paired-end reads against SPB-specific sequences within the *rfb* cluster (Supplementary Table 2) and *fliC* (encoding the H1 flagellin) and *fljB* (encoding the H2 flagellin), with calling by the NCBI BLAST + [63] v.2.14.1 blastn command line tool on assemblies against SPB reference sequences (Supplementary Table 2). The specific STM 3356 SNV present in *d*-Tar⁻ isolates[22] was sought with the NCBI BLAST+ v.2.14.1 blastn command line tool on assemblies against SPB⁻ and SPB⁺ reference sequences (Supplementary Table 2).

Multilocus sequence typing (MLST)[21], and core genome MLST (cgMLST) were performed with various tools integrated into EnteroBase[64] (https://enterobase.warwick.ac.uk/species/index/senterica). The EnteroBase *Salmonella* cgMLST scheme ("cgMLST V2 + HierCC V1") − based on 3002 core genes − assigns bacterial genomes to single-linkage hierarchical clusters (HCs) at 13 fixed levels of resolution, ranging from HC0 (high-resolution clusters consisting of identical genomes with no allelic differences) to HC2850 (low-resolution clusters consisting of genomes with up to 2850 allelic differences). Evaluations by Zhou and coworkers[65] found that, in the genus *Salmonella*, cluster assignments at the HC2850, HC2000, and HC900 levels could be used to distinguish subspecies, super-lineages and eBurst groups[21], respectively, and that epidemic outbreaks could be distinguished with HC2, HC5, or HC10. Sequence type and cgMLST clustering results at the HC2, HC5, HC10, HC20, HC50, HC100, HC200, HC400, HC900, HC2000, HC2600 and HC2850 levels are shown in Supplementary Data 1 for the genomes from the diversity dataset and in Supplementary Data 6 for the genomes from the surveillance dataset.

We also constructed a cgMLST tree − based on core genome allelic distances −inferred with the NINJA neighbour-joining algorithm (present in the "cgMLST V1+HierCC V1" scheme) and visualised with GrapeTree[66] for two different datasets. The first dataset corresponded to the 166 genomes from Connor and coworkers[27] present in EnteroBase (https://enterobase.warwick.ac.uk/species/senterica/search_strains?query=workspace:86472), as 12 genomes did not pass the quality control of EnteroBase and one was discarded due to a probable inversion (Supplementary Data 7). The second corresponded to the 567 genomes from our diversity dataset that were present in EnteroBase (https://enterobase.warwick.ac.uk/species/senterica/search_strains?query=workspace:86468); one unpublished draft genome (ATCC 10719, SRR955214) sequenced with 454 technology could not be uploaded by EnteroBase, which accepts only complete genomes or Illumina short reads. The resulting cgMLST trees used for Supplementary Figs. 1 and 6 are publicly available from https://enterobase.warwick.ac.uk/ms_tree?tree_id=92077 and https://enterobase.warwick.ac.uk/ms_tree?tree_id=92095, respectively. We found that some of the strains described in the Supplementary Table 1 of the article by Connor et al.[27] were assigned to incorrect STs, PGs (Supplementary Data 7). We have corrected the assignments for this study.

EnteroBase was searched on April 9th 2021, with the HC400_1620 criterion (see Supplementary Note "Validation of the SPB⁻ PG1 diversity dataset") used to identify additional genomes, thereby extending the geographic and time coverage of our diversity dataset. EnteroBase was also searched on April 10th 2023 with the same HC400_1620 criterion, to assemble the surveillance dataset of 336 genomes (https://enterobase.warwick.ac.uk/species/senterica/search_strains?query=workspace:91222).

## Detection of antimicrobial resistance determinants

Antimicrobial resistance (AMR) genes were detected with SRST2 (ref. [67]) v.0.2.0 using default parameters and the CARD[68] v.3.0.8 AMR gene database. We also determined the presence or absence of mutations in the quinolone resistance-determining regions (QRDRs) by extracting the relevant SNV calls for codons 83 and 87 of *gyrA*, codon 464 of *gyrB*, and codon 80 of *parC*, from the SPB⁻ PG1 reference genome CIP 54.115 (GenBank accession no. CP147898) (Supplementary Data 2). We searched for AMR plasmids with PlasmidFinder[69] v.2.0 (https://cge.food.dtu.dk/services/PlasmidFinder-2.0/).

## Phylogenetic analyses

The paired-end reads and simulated paired-end reads were mapped onto the complete SPB⁻ PG1 reference genome CIP 54.115 (GenBank accession no. CP147898) with RedDog (https://github.com/katholt/reddog-nf). Genomes with a depth of ≤10x across the reference genome and a ratio of heterozygous SNVs/homozygous SNVs > 0.7 were excluded from the analysis, leaving 568 genomes for downstream analysis. We excluded SNVs present in fewer than 99% of genomes, and SNVs in repeat regions, such as insertion sequences or phages. Recombination was masked with Gubbins[70] v.2.3.2 using the weighted_robinson_foulds parameter. The resulting SNV alignment of 15,995 SNVs (see Supplementary Data 2 for a full list of locations) was used to create a maximum likelihood phylogeny with RAxML[71] v.8.2.9, with a GTR + G base substitution model and 1000 bootstraps. The final tree was rooted on the lineage L1 SPB⁻ PG1 genome NCTC 8299 and visualised with iTOL[72] v.6 (https://itol.embl.de).

We first assessed the temporal signal in the diversity dataset (with and without the five outliers) by "root-to-tip" regression (linear regression of the number of substitutions accumulated from the root to the tips of the ML phylogenetic tree as a function of sampling times) with TempEst[73] v.1.5.3 (Supplementary Fig. 3 and Supplementary Data 1). We confirmed the temporal signal in a subset of 256 genomes (see below) by performing eight date randomisations[74] in BEAST[75] v.2.7.1. The eight runs with randomised dates gave significantly different estimates for substitution rate compared to the run with real dates (Supplementary Fig. 3), indicating a strong temporal signal in the real data.

For the generation of a timed phylogeny, we selected a subset of 256 genomes representing the diversity of the phylogeny and the full range of isolation dates and geographies (Supplementary Data 1). We used all genomes in lineages L1 to L6 and L8, and lineages L7, L9 and L10 were subsampled by selecting the oldest and newest genomes in each lineage, together with a random selection of the remaining genomes in the lineage, keeping the numbers of genomes sampled proportional to the full dataset (Supplementary Data 1). We used BEAST v.2.7.1 with a GTR substitution model, the optimised relaxed clock model (setting the ucldMean prior to a lognormal distribution with a mean of 0.0001 and a standard deviation of 2, an initial value of 0.0001, and ensuring the 'mean in real space' option was checked) and the Bayesian Skyline population model. We performed three independent runs of 600 million states. We removed a 10% burn-in for each run and combined the runs together to create the consensus file. A consensus tree was drawn with the 'majority rule' option from sumtrees.py in dendropy[76] v.4.5.2, setting branch lengths to the median length. The skyline plot was drawn with Tracer[77] v.1.7.2.

## Defining lineages and genotypes

We used fastbaps[78] v.1.0.8 and visual inspection to define lineages in the maximum likelihood tree, using the best_baps_partition function with the phylogeny as a prior.

Genotypes within lineages were defined following visual inspection. For the selection of marker SNVs for each genotype, all SNVs were mapped onto the phylogeny with SNPPar[79] v.1.1. On branches leading to genotypes, we selected marker SNVs, prioritising SNVs that were synonymous mutations within genes for which the ratio of non-synonymous SNVs to synonymous SNVs was as close to zero as possible. The final selection of marker SNVs, their coding consequences and genome location relative to the reference can be found in Supplementary Data 5.

## Implementation and validation of the genotyping scheme

We implemented the genotyping scheme in Mykrobe[80] v.0.12.2. The scheme is therefore supported by this version onwards and is installed when running the Mykrobe commands "panels update_metadata" and "panels update_species all". We used the pre-existing *invA* probe[81] to detect genomes belonging to *S. enterica*. An additional probe was created to distinguish between SPB PG1 genomes and other serotypes, including non-PG1 Paratyphi B genomes − this probe detects the *d*-Tar-specific SNV (i.e., a G- > A change in the start codon of STM 3356)[22] (Supplementary Table 2). We tested Mykrobe implementation on all 568 genomes from the diversity dataset (Supplementary Data 1), using Illumina reads as input, to confirm genotypes were assigned correctly. We then tested the scheme on the 336 genomes of the surveillance dataset (Supplementary Data 6). All these genomes were mapped onto the reference genome as described above and included in the ML phylogeny (which was constructed with IQTree[82] v.2, with a GTR substitution model, on 17,338 SNVs), resulting in 793 genomes (Supplementary Fig. 11). We assigned a genotype to each genome with Mykrobe, and genotypes were validated against the phylogeny to ensure they were correct.

## De novo assembly

Assemblies were generated from Illumina paired-end reads with the fq2dna/21.06 script (https://gitlab.pasteur.fr/GIPhy/fq2dna strategy B; default settings).

Assemblies were generated from 454-GS-FLX reads with SPAdes[83] v.3.15.5 (k-values of 21, 37, 53, 69, 77, default settings for other criteria).

## Pan-genome analysis

The pan-genome analysis was performed on the diversity dataset, which included 566 Illumina paired-end reads, one published complete genome (P7704), and one publicly available draft genome from 454-GS-FLX reads (CFSAN024725) (Supplementary Data 1). We used panaroo[84] v.1.3.0, with default parameters and genome assemblies annotated with bakta[61] v.1.5.0. We estimated the pan-genome openness level (open if alpha ≤ 1 and closed if alpha is > 1) for the 568 genomes, using Heaps' law with the R micropan[85] v.2.1 package. The gene presence/absence matrix of the pan-genome created is shown in Supplementary Data 3. We increased the accuracy of accessory gene assignment to prophages or plasmids by performing a second pan-genome analysis on 581 genomes constructed from 567 short-read assemblies and 14 complete genomes (see next paragraph). After manual curation of the prophage content in the complete genomes (see next paragraph), we selected the 1,506 accessory gene IDs of the 568-pan-genome in the 581-pan-genome output, and we were able to assign the 1,506 accessory CDS to prophage and plasmid regions. An additional manual curation of short-read assemblies was performed for plasmid contigs potentially absent from the 14 complete genomes (Supplementary Data 4).

## Prophage content analysis and *sopE* copy-number variation

Unlike short-read assemblies, complete genome assemblies provide complete information about prophage content and location. We therefore used 14 complete SPB⁻ genomes: the genome included in our diversity dataset (P7704), and 13 additional genomes from isolates with short reads included in the "diversity dataset": SARA41_FB_1, which was publicly available (GenBank accession no. CP074225.1) and 12 genomes newly generated for this study (Materials section "Long-read sequencing and complete genome circularisation")

(Supplementary Data 1). The prophage regions were detected in these genomes and taxonomically classified with the PHASTER[86] tool in June 2023. Prophages annotated as intact and absent in at least one of the 14 complete genomes were retained for further analysis. Ten prophage regions were then precisely delineated from the alignment of the 14 genomes around each insertion site (Supplementary Data 8). Finally, we screened for occupancy of the 10 insertion sites in the 568 genomes of the diversity dataset with the blastn algorithm (BLAST + [63] v.2.14.1).

By combining the annotation of the 14 SPB⁻ complete genomes and the prophage delineation described above, we were able to locate the *sopE* gene in SEN34-like and P88-like prophages. We used the blastn algorithm and the *sopE* reference sequence (GenBank accession no. L78932) from *S. enterica* serotype Dublin[37] to confirm our identification of the prophage-borne *sopE* gene.

As the number of *sopE*-containing prophages differed between genomes, we obtained a good estimate of the *sopE* prophage copy number per genome (*sopE*-CN) by combining the insertion site occupancy and short-read coverage data. The short reads were first aligned with the complete genome of the B62 isolate (GenBank accession no. CP147902), with the very-sensitive-local option of bowtie2 (ref. [87]) v.2.3.5.1, and coverage was estimated across the entire genome (cov_g), at the *sopE* locus (cov_s) and the *Salmonella* Pathogenicity Island 1 locus (GenBank accession no. CP147902; coordinates 981997 – 1036581) (cov_pi) with samtools[88] v.1.13. The mean value of the cov_s/ cov_pi and cov_s/ cov_g ratios was chosen as an estimate of the *sopE*-CN. The sequencing platform has a strong and significant effect on *sopE*-CN values (Kruskal-Wallis test $p$ = 4.8E-50, large magnitude effect eta2[H] = 0.403). We therefore split the dataset according to the Illumina platform used (HiSeq, $n$ = 274; MiSeq, $n$ = 45; or NextSeq, $n$ = 242) before searching for outliers in the *sopE*-CN distribution with Dunn's pairwise comparison test with Bonferroni correction (Supplementary Fig. 12). If the *sopE*-CN and the number of insertion sites occupied were not correlated, we visually inspected both the long- and short-read mappings onto the genomes of isolates B62 and B2590, with Tablet[89] v.1.21.02.08. The complete data are summarised in Supplementary Data 9.

Prophages inserted in the vicinity of the *hin-fljB-fljA* cluster are described in Supplementary Data 8. The assemblies from the 568 diversity dataset genomes were screened for the presence and structure of the *hin-fljB-fljA* gene cluster with the blastn[90] algorithm.

### Structure of the *hin-fljB-fljA* gene cluster
We used the gggenes[91] v.0.5.0 and ggplot2 (ref. [92]) v.3.4.2 packages of R[93] v.4.1.2 to visualise prophage gene content.

### Statistics
Statistical analysis was performed with R v.4.1.2, R packages rstatix[94] v.0.7.2, and ggpubr[95] v.0.6.0.

### Data collection
The data were entered into an Excel (Microsoft) version 16.76 spreadsheet or tabulation-separated text files (tsv).

### Reporting summary
Further information on research design is available in the Nature Portfolio Reporting Summary linked to this article.

### Data availability
The publicly available sequences used in this study are available from GenBank (https://www.ncbi.nlm.nih.gov/genbank/) under accession numbers NC_028699, NC_026014, NC_010102.1, CP074225.1, L78932, NZ_CP065185, NZ_CP065186.1, NZ_CP065187.1, NZ_CP065188.1, and JWQX00000000.1.

The long-read sequence data generated in this study are available from GenBank under accession numbers CP147895, CP147896,

CP147897, CP147898, CP147899, CP147900, CP147901, CP147902, CP147903, CP147904, CP147905, CP147906, and CP147907. The short-read sequence data generated in this study were submitted to EnteroBase (https://enterobase.warwick.ac.uk/) and to the European Nucleotide Archive (ENA, https://www.ebi.ac.uk/ena/) under study numbers PRJDB11608, PRJEB18998, PRJEB28356, PRJEB30317, PRJEB67705, PRJNA248792, PRJEB68323, PRJEB49424, PRJEB71958. All the accession numbers of the short-read sequences produced and/or used in this study are listed in Supplementary Data 1 and Supplementary Data 6.

The list of genomes studied (and their assembled short-read data) can be obtained from EnteroBase at: https://enterobase.warwick.ac. uk/species/senterica/search_strains?query=workspace:86468 (diversity dataset, Supplementary Data 1) and: https://enterobase.warwick. ac.uk/species/senterica/search_strains?query=workspace:91222 (surveillance data set, Supplementary Data 6).

The list of 166 genomes (and their assembled short-read data) published by Connor et al.[27] and present in EnteroBase can be obtained from https://enterobase.warwick.ac.uk/species/senterica/search_ strains?query=workspace:86472 (Supplementary Data 7).

The cgMLST GrapeTrees shown in Supplementary Figs. 1 and 6 can be visualised with EnteroBase at: https://enterobase.warwick.ac. uk/ms_tree?tree_id=92077 and https://enterobase.warwick.ac.uk/ms_ tree?tree_id=92095, respectively. Source data are provided with this paper.

### Code availability
Mykrobe is available for download at https://github.com//mykrobe-tools/mykrobe. Instructions for running Mykrobe for SPB⁻ PG1 are available in the Mykrobe documentation at https://github.com/ Mykrobe-tools/mykrobe/wiki/AMR-Prediction. Briefly, the option --species paratyphiB should be used when running the Mykrobe predict command. The genotyping panel developed here is available from https://figshare.com/articles/dataset/Mykrobe_panel_paratyphi_B_ version_20230627/24925506

The fq2dna script (genome de novo assembly from raw paired-end FASTQ files) can be found at https://gitlab.pasteur.fr/GIPhy/ fq2dna.

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

## Acknowledgements

We thank Prof. Jacques Ravel, Prof. David A. Rasko, Luke Tallon, Kranthi Vavikolanu, and Michael Pietsch for submitting archived or new short reads to a public repository, Susan Van Duyne for ensuring the safe shipping of strains, Anthony M. Smith and Chien-Shun Chiou for reviewing their data, Paul O'Dette for his support, and the sequencing team at the *Institut Pasteur* (PF1 & P2M-*Plateforme de Microbiologie Mutualisée*) for sequencing the samples. We also thank all the corresponding laboratories of the French National Reference Centre for *Escherichia coli*, *Shigella*, and *Salmonella*.

This research was funded by the *Fondation Le Roch-Les Mousquetaires* (to F.-X.W); *Institut Pasteur* (to F.-X.W); *Santé publique France* (to F.-X.W); and by the French Government's *Investissement d'Avenir* programme, *Laboratoire d'Excellence* 'Integrative Biology of Emerging Infectious Diseases' (grant no. ANR-10-LABX-62-IBEID to F.-X.W). M.H. was funded by the National Institute for Health Research Health Protection Research Unit (NIHR HPRU) in Healthcare Associated Infections and Antimicrobial Resistance at Oxford University in partnership with the UK Health Security Agency (NIHR200915), and the NIHR Biomedical Research Centre, Oxford. M.A.C. is affiliated to the National Institute for Health Research Health Protection Research Unit (NIHR HPRU) (NIHR200892) in Genomics and Enabling Data at University of Warwick in partnership with the UK Health Security Agency (UKHSA), in collaboration with Universities of Cambridge and Oxford. M.A.C. is based at UKHSA. The views expressed are those of the authors and not necessarily those of the Centers for Disease Control and Prevention, the NIHR, the Department of Health and Social Care or the UK Health Security Agency. The funders had no role in study design, data collection and analysis, decision to publish, or preparation of the manuscript.

## Author contributions

F.-X.W. designed and oversaw the study. D.B., M.A.C., S.S., H.I., P.I.F., N.d.L., L.K., X.X., J.I., D.C., Y.A., M.MA., Y.W., B.M.J., Z.M., M.C., M.Y., B.Z., M.MO., C.S.N., M.P.G., and F.-X.W. selected and provided isolates or genomes with their basic metadata. L.FR., A.T.D., S.I.J., M.R., V.G., E.S., and L.FA. subcultured the bacteria, performed phenotypic experiments, and extracted DNA. J.H., L.FR., A.T.D., A.Z., E.N., L.FA., and F.-X.W. analysed and/or interpreted the data. J.H., M.H. and Z.I. implemented the genotyping scheme in Mykrobe. J.H., L.FR., and F.-X.W. wrote the manuscript. All the authors contributed to the editing of the manuscript.

## Competing interests

The authors declare no competing interests.

## Additional information

[1]Department of Infectious Diseases, School of Translational Medicine, Monash University, Melbourne, VIC 3004, Australia. [2]Institut Pasteur, Université Paris Cité, Unité des Bactéries pathogènes entériques, Paris F-75015, France. [3]Institut Pasteur, Université Paris Cité, Bioinformatics and Biostatistics Hub, Paris F-75015, France. [4]Scottish Microbiology Reference Laboratories (SMiRL), Glasgow G31 2ER, UK. [5]Gastrointestinal Bacteria Reference Unit (GBRU), United Kingdom Health Security Agency, London NW9 5EQ, UK. [6]Unit of Enteropathogenic Bacteria and Legionella (FG11)/National Reference Centre for Salmonella and Other Bacterial Enteric Pathogens, Robert Koch-Institute, Wernigerode 38855, Germany. [7]Department of Bacteriology I, National Institute of Infectious Diseases, Tokyo 162-8640, Japan. [8]Division of Foodborne, Waterborne and Environmental Diseases, Centers for Disease Control and Prevention, Atlanta, GA, USA. [9]National Salmonella, Shigella and Listeria Reference Laboratory, Galway University Hospitals, Galway SW4 671, Ireland. [10]Pasteur Institute of St Petersburg, St Petersburg 197101, Russia. [11]Department of Microbiology, Shanghai Municipal Centre for Disease Control and Prevention, Shanghai 200336, China. [12]Department of Bacteriology, Toyama Institute of Health, Toyama 939-0363, Japan. [13]Institut Pasteur, Université Paris Cité, Collection of Institut Pasteur (CIP), Paris F-75015, France. [14]International Joint Research Centre for National Animal Immunology, College of Veterinary Medicine, Henan Agricultural University, Zhengzhou, Henan 450046, China. [15]CAS Key Laboratory of Pathogen Microbiology and Immunology, Institute of Microbiology, Chinese Academy of Sciences (CAS), Beijing 100101, China. [16]European Molecular Biology Laboratory, European Bioinformatics Institute, Hinxton CB10 1SD, UK. [17]Nuffield Department of Medicine, University of Oxford, Oxford, UK. [18]National Institute of Health Research Oxford Biomedical Research Centre, John Radcliffe Hospital, Headley Way, Oxford, UK. [19]Health Protection Research Unit in Healthcare Associated Infections and Antimicrobial Resistance, University of Oxford, Oxford, UK. [20]School of Medicine, University of Galway, Galway H91 TK33, Ireland. [21]Department of Veterinary Medicine, Zhejiang University College of Animal Sciences, Hangzhou 310058, China. [22]School of Life Science, Hangzhou Institute for Advanced Study, University of Chinese Academy of Sciences, Hangzhou 310024, China. [23]Milner Centre for Evolution, University of Bath, Claverton Down, Bath, UK. [24]Present address: Bioinformatic Core Facility, UMS AMMICA, Gustave Roussy, Villejuif F-94800, France. [25]These authors contributed equally: Jane Hawkey, Lise Frézal. ✉e-mail: francois-xavier.weill@pasteur.fr

