## [Peer Review File · Nature Communications]

REVIEWER COMMENTS

Reviewer #1 (Remarks to the Author):

The authors present compelling analyses documenting the genetic diversity, population structure, and historical distribution from a well-curated dataset of *S. Paratyphi B d-tartrate(-)* strains. These results provide novel insight into the global distribution of SPB(-), which continues to be an important public health concern. Furthermore, informed by their genomic analyses, the authors developed a pipeline for rapid and accurate typing of SPB(-) strains for facilitating public health surveillance efforts. The data analysis, interpretation and conclusions are sound. There are some speculations about the historical events that could have led to the observed distribution in SPB, but these points are brought up in the discussions section of the paper.

Overall, this manuscript is based on compelling, sound, and robust evidence. I have only minor suggestions for the authors to consider.

Line 116: what is synthetic cream? Non-dairy cream?

Lines 200-201: How were the higher-resolution genotypes named? Some have locations associated with their names? Since attaching geographic locations to names has some important public health implications, more details about % of isolates from a given geographic location to assign clade name would be helpful to include, especially in the cases where the groups have multiple geo locations (ex. NorthAfrica_Europe).

Line 319: I might have missed this, but how was the presence of *sopE* confirmed/determined? With *blastn*? Or was this denoted from the Phaster annotation? I think this is important to describe because the authors present support for using *sopE* as a marker for SPB- strains (line 455)

Lines 334-337: Were the sequences of *sopE* at the three genomic insertion sites different? Was *sopE* at insertion site 1 distinguishable from *sopE* at insertion site 2 or 3? The analyses suggest that they may be carried on different prophages.

Line 430: add 'in recent years' – it is noted at the beginning of the section that the surveillance dataset is used for this conclusion. Also, the three reasons listed are the 'primary' reasons

Line 490: Why 10 lineages? Earlier in the paper (Fig. 1) the authors note the presence of 11 lineages including 3 with singletons, of which lineage 11 is one

Reviewer #2 (Remarks to the Author):

Thank you for the opportunity to review this manuscript which describes the population diversity and evolution of *Salmonella enterica* serotype Paratyphi B or SPB. This organism of one of four causes of

enteric fever, a global public health challenge for many centuries.

This is an original, well-planned and well-presented study by experts in microbiology of enteropathogens.

The authors focused largely on D-tartrate non-fermenting strains of SPB and examined the population structure and temporal evolution of 568 genomes, the majority of which were generated for this study. The study identified 11 lineages (with the predominance of L10) and 38 SNVs unique to each SPB genotype and proposed and implemented a hierarchical SNV-based genotyping scheme that can split SPB populations into phylogeographically informative genotypes. The core genome of 4,044 genes was identified and proportions of accessory genes belonging to prophages, plasmids and transposases were quantified.

They also assembled and investigated a set of 336 genomes from four major public health laboratories in North America and Europe, which were uploaded between 2015 and 2023. The evolutionary timescale analysis and Bayesian analysis of population structure was applied to the data.

The paper builds on a rich history of discovery of salmonellas associated with clinical typhoid infections, fills a current gap in our understanding of the genomic epidemiology of this pathogen and should be of interest to readers of the Journal. The methods employed in this study are cutting-edge in microbial genomics and are adequate for the research aims and hypotheses.

In order to further improve the paper, the authors may consider addressing the following suggestions and questions:

- How informative could the suggested SPB genotyping system be for the public health investigation of outbreaks? What does the surveillance dataset tell us in that respect?
- Similar population genomics studies of *Salmonella Paratyphi A* emphasised the role gene acquisitions or losses play as key molecular events in the evolution of new lineages (e.g., Jacob JJ et al. Genomic analysis unveils genome degradation events and gene flux in the emergence and persistence of *S. Paratyphi A* lineages. *PLoS Pathogens* 2023;19(4): e1010650). Have the authors observed any parallels in their studies of SPB?
- Clarify whether any SPB isolates from Connor's collection (Connor TR et al. *mBio* 2016; 7(4): e00527-16 or Reference 27) were included in the study dataset of 568 genomes?
- Please provide percentages for cases where quinolone resistance conferring mutations were documented (lines 294-298).
- The important observation of azithromycin resistance presented in Figure 5 deserves to be mentioned in the text with the mechanism of resistance explained. The term 'azithromycin' (i.e. an individual antibiotic name) in Figure 5 can be replaced with 'macrolides' (antibiotic class name) to be consistent with other terms in the legend.
- Table 2 – Suggested heading for the second column – “Years of MRCA (95% HPD)”.
- Some minor editorial suggestions:
 - o Rephrase the statement in the Abstract “We show that this pathogen existed in the 13th century” to “Our comparative genomics findings suggest that this pathogen has existed since the 13th century”.
 - o Change “exploitation’ to ‘exploration’ (line 391), ‘their colonial and immigration histories’ (line 431) to ‘migration and travel patterns over recent centuries’
 - o Change to ‘Universities’ (line 544).

Reviewer #3 (Remarks to the Author):

This is an interesting and very detailed study of the lineage of Salmonella that is the causative agent of Paratyphi B. It is well written, and I enjoyed reading it. It is certainly worthy of publication and will be of broad interest.

Novelty

I think the paper has broad appeal, both within the field and in related fields. The work itself is original, and also builds well on the existing literature. As well as bringing together and drawing a line under findings from other studies (including those by the authors and others), the paper includes some novel/important findings. Firstly, the detailed historical collection shows strong evidence of phylogeographic clustering, which is not something that has been reported before, and is of epidemiological value and scientific value. Secondly, the pattern of resistance is also interesting, including evidence for increasing resistance in SPB isolates recently. Thirdly, The work around the prophages within SPB is also novel, and the pangenome analysis is also more comprehensive than previous work, due to a combination of the larger historic dataset and the use of long read technologies. Lastly, the observation around S. Onarimon, in the supplementary is also of interest to those who are enthusiasts of the Salmonellae, and it was an unexpected nugget that was also new and interesting.

Support for claims

Overall, the paper has a large amount of detail, and represents a very in-depth genomic study of this interesting pathogen.

The paper brings together historical work very well along with a historical and contemporary genomic dataset to provide the highest resolution exploration to date of SPB. The approaches used are appropriate and well matched to the aims, and the methods, both laboratory and bioinformatics are explained in sufficient detail to enable reproducibility. The use of Enterobase is welcome and enhances the reproducibility of the work. The release of the probes for Mykrobe is also welcome as this is open-source software.

Overall, the data analysis is sensible, and the methods are appropriate and meet the expected standards in the field.

Other questions/comments for the authors:

Although I like this paper a lot and think it should be published, there are a number of questions/clarifications that would be helpful, these are;

The paper reports the correlation with geography and the evidence of geographic clustering; however, I couldn't see a formal analysis for the phylogeography. Given that the authors had made use of BEAST, I wondered if they had made use of any of the phylogeographic models to examine predicted origins and phylogeographic spread more formally.

So

1. Were analyses performed to substantiate the suggestion of export from Europe?
2. The detail around the BEAST analysis is in the supplementary, and these results are interesting, although broadly agreeing with what had been found before - but the BEAST date/context information isn't that well referenced in the main text - so could the authors consider how to better signpost to the detail in the supplementary information regarding BEAST.

The AMR analysis is interesting, but it wasn't clear to me from the paper in what context the resistance genes were found. So;

3. Were the genes found on chromosomal mobile elements, carried with/on phage or were they plasmid carried?
4. From the figure in the supplementary, it looks like there weren't many plasmids in the samples analysed, though it would be good to have this described in a little more detail. The paper mentions a phylogenetic comparison between plasmid genes and core genome, but it isn't clear to me what this means, or what it shows - can the authors explain what is meant and consider if figure 9 in the supplementary is appropriate to communicate this.

With respect to the phylogenetics and cgMLST analyses;

5. How many (if any) HC400_1620 samples in Enterobase did not contain the STM 336 SNV? It isn't wholly clear from the text if HC400_1620 is just SPB- strains, or if it includes both SPB+ and SPB- strains.
7. Did the Onarimon samples cluster in a single lineage? were they grouped with other more contemporary samples in a single lineage? it isn't clear from the paper/results where these sit with respect to other lineages within SPB- samples.

Lastly;

8. In the validation work around Mykrobe, was the probe tested against a wider database to identify if the probe could miss-detect SPB? what validation was done on this - as the MS only lists testing against the study datasets.

9. In the discussion there are assertions around long term carriage in elderly patients - it wasn't clear to me what the evidence was for this. In those contemporaneous samples from older patients, was there evidence of carriage (e.g., diversity) or long-term chronic infection in their genomes?

10. Were the 'carriage' isolates resistant to antibiotics (which one might have expected - as one would have expected some antibiotic exposure in older patients at some point, which you might have expected to impact carriage of SPB)?

Authors: We would like to thank the reviewers for their many insightful comments. We address these comments, point by point, below.

REVIEWER COMMENTS

Reviewer #1 (Remarks to the Author):

The authors present compelling analyses documenting the genetic diversity, population structure, and historical distribution from a well-curated dataset of *S. Paratyphi B* d-tartrate(-) strains. These results provide novel insight into the global distribution of SPB(-), which continues to be an important public health concern. Furthermore, informed by their genomic analyses, the authors developed a pipeline for rapid and accurate typing of SPB(-) strains for facilitating public health surveillance efforts. The data analysis, interpretation and conclusions are sound. There are some speculations about the historical events that could have led to the observed distribution in SPB, but these points are brought up in the discussions section of the paper.

Overall, this manuscript is based on compelling, sound, and robust evidence. I have only minor suggestions for the authors to consider.

Line 116: what is synthetic cream? Non-dairy cream?

Authors: It is an emulsion of vegetable oils or fat with water, with or without the addition of other substances, some of which may be of dairy origin. It was used in the UK during World War II when the sale of natural cream was forbidden. This information has been added to the Supplementary Note “Epidemiology of paratyphoid B fever during the first half of the 20th century”.

Lines 200-201: How were the higher-resolution genotypes named? Some have locations associated with their names? Since attaching geographic locations to names has some important public health implications, more details about % of isolates from a given geographic location to assign clade name would be helpful to include, especially in the cases where the groups have multiple geo locations (ex. NorthAfrica_Europe).

Authors: We now indicate in the results section: “*Strong geographic patterns with differences from country to continent level were observed for 17 genotypes, whereas two genotypes were more widespread, isolated from two continents (Supplementary Data 1). These two genotypes were genotype 7.2 found in Europe (26.3%, 5/19) and Asia (73.7%, 14/19, mostly East Asia), and genotype 10.3.8.5 found in Europe (71.4%, 20/28) and North Africa (25%, 7/28). One genotype, 7.3.2 — also found in Europe (41.2%, 7/17) and North Africa (52.9%, 9/17) — was associated with a particular PT, BAOR (see Comparison of phylogenomics data with other typing schemes) (Supplementary Data 1). This geographic or PT information was added to the genotype nomenclature as an alias, to make it more informative.*” and in the discussion section: “*New genotypes with geographic information (in cases of new emerging genotypes in a defined area) or new updates on alias names (in cases of the establishment of known genotypes in new areas) should be added to the scheme in the future.*”

Line 319: I might have missed this, but how was the presence of sopE confirmed/determined? With blastn? Or was this denoted from the Phaster annotation? I

think this is important to describe because the authors present support for using *sopE* as a marker for SPB- strains (line 455)

Authors: To clarify this point, we have added the following sentences to the M&M section entitled “Prophage content analysis and *sopE* copy-number variation”: “By combining the annotation of the 14 SPB⁻ complete genomes and the prophage delineation described above, we were able to locate the *sopE* gene in SEN34-like and P88-like prophages. We used the *blastn* algorithm and the *sopE* reference sequence (GenBank accession no. L78932) from *S. enterica* serotype Dublin³⁸ to confirm our identification of the prophage-borne *sopE* gene.”

Lines 334-337: Were the sequences of *sopE* at the three genomic insertion sites different? Was *sopE* at insertion site 1 distinguishable from *sopE* at insertion site 2 or 3? The analyses suggest that they may be carried on different prophages.

Authors: The *sopE* gene nucleotide sequence is highly conserved across prophages. We now indicate the following in results section: “In the 14 complete genomes, the nucleotide sequence of the *sopE* gene was 100% identical, regardless of the prophages in which it is inserted. One exception was the *sopE* gene inserted at site #3 of the B624 genome, which differed by eight (five being non-synonymous) out of 723 nucleotides from the *sopE* consensus sequence of SPB⁻.”

Line 430: add ‘in recent years’ – it is noted at the beginning of the section that the surveillance dataset is used for this conclusion. Also, the three reasons listed are the ‘primary’ reasons

Authors: This has been done.

Line 490: Why 10 lineages? Earlier in the paper (Fig. 1) the authors note the presence of 11 lineages including 3 with singletons, of which lineage 11 is one

Authors: We were actually referring to the 10 lineages or phylogroups described by Connor et al for the global population SPB and not the lineages we have identified for SPB PG1. We have clarified this in the following sentence: “Firstly, SPB isolates can be assigned to the 10 known PGs described by Connor and coworkers²⁷ with the Enterobase cgMLST scheme, with the HC400_1620 cluster considered a signature of SPB⁻ PG1.”.

Reviewer #2 (Remarks to the Author):

Thank you for the opportunity to review this manuscript which describes the population diversity and evolution of *Salmonella enterica* serotype Paratyphi B or SPB. This organism of one of four causes of enteric fever, a global public health challenge for many centuries. This is an original, well-planned and well-presented study by experts in microbiology of enteropathogens.

The authors focused largely on D-tartrate non-fermenting strains of SPB and examined the population structure and temporal evolution of 568 genomes, the majority of which were generated for this study. The study identified 11 lineages (with the predominance of L10) and 38 SNVs unique to each SPB genotype and proposed and implemented a hierarchical SNV-based genotyping scheme that can split SPB populations into phylogeographically informative genotypes. The core genome of 4,044 genes was identified and proportions of accessory genes belonging to prophages, plasmids and transposases were quantified.

They also assembled and investigated a set of 336 genomes from four major public health

laboratories in North America and Europe, which were uploaded between 2015 and 2023. The evolutionary timescale analysis and Bayesian analysis of population structure was applied to the data.

The paper builds on a rich history of discovery of salmonellas associated with clinical typhoid infections, fills a current gap in our understanding of the genomic epidemiology of this pathogen and should be of interest to readers of the Journal. The methods employed in this study are cutting-edge in microbial genomics and are adequate for the research aims and hypotheses.

In order to further improve the paper, the authors may consider addressing the following suggestions and questions:

- How informative could the suggested SPB genotyping system be for the public health investigation of outbreaks? What does the surveillance dataset tell us in that respect?

Authors: We now expand on this aspect in the revised MS (discussion section). The text now reads: *“We anticipate that the use of this scheme and its universal nomenclature will improve the laboratory surveillance of PTB. It is now easier to track the different SPB⁻ PG1 populations at global scale. We were able to identify the different genotypes detected recently in travellers from or migrants to four high-income countries in North America and Europe (surveillance dataset). However, more global studies involving countries experiencing PTB across the globe and performed regularly over time — as for the PT surveys — would be helpful to monitor the diversity, spread and evolution (of AMR in particular) of this pathogen. New genotypes with geographic information (in cases of new emerging genotypes in a defined area) or new updates on alias names (in cases of the establishment of known genotypes in new areas) should be added to the scheme in the future. The use of a common nomenclature would also be helpful during outbreak investigations. It is now straightforward to define the bacterial types of SPB⁻ PG1 and to share this information during transborder outbreak investigations. This should also facilitate the identification of transmission chains associated with particular geographic regions, particularly in areas in which PTB is not endemic. For example, genomic surveillance in the UK identified an imported SPB⁻ outbreak in travellers coinciding with a mass gathering in Iraq in 2021 (ref.²⁹). The isolates from these patients clustered in one of the two clades labelled “travel to Iraq”. According to our genotyping scheme, this clade corresponds to genotype 10.3.2_MiddleEast1. The oldest isolates belonging to this genotype were collected in Iran in 1965 and Iraq in 1975, suggesting that this strain has been endemic in the region for many decades. The second clade labelled “travel to Iraq” identified by UKHSA corresponded to genotype 10.3.8.3_MiddleEast3. In endemic regions, this genotyping scheme might also be useful for determining the genotype of a potential outbreak strain if several genotypes are known to circulate regionally.”*

- Similar population genomics studies of Salmonella Paratyphi A emphasised the role gene acquisitions or losses play as key molecular events in the evolution of new lineages (e.g., Jacob JJ et al. Genomic analysis unveils genome degradation events and gene flux in the emergence and persistence of S. Paratyphi A lineages. PLoS Pathogens 2023;19(4): e1010650). Have the authors observed any parallels in their studies of SPB?

Authors: This interesting aspect of genome degradation events and gene flux leading to the emergence of SPB PG1, a human-adapted pathogen causing paratyphoid fever, was previously studied by Connor et al (our reference #27) in their comparative genomics

analysis of the different PGs (1-10) of SPB. We did not address this question as our study focused exclusively on the PG1 population. Regarding the evolution of PG1, our comprehensive pangenome analysis failed to identify genomic signals leading to increases in virulence or transmissibility during recent microevolution (as previously reported for other enteric fever pathogens, such as *S. enterica* serovars Typhi (Roumagnac et al. Science 2016) and Paratyphi A (Zhou et al. PNAS 2014)). Regarding virulence, we documented up to three copies of the *sopE* gene in some lineages, but we have no other data linking this microbial genomics finding to an increase in virulence *in vivo* or in human populations.

- Clarify whether any SPB isolates from Connor’s collection (Connor TR et al. mBio 2016; 7(4): e00527-16 or Reference 27) were included in the study dataset of 568 genomes?
Authors: The section entitled “*S. enterica* Paratyphi B sequence data collections” of the M&M (lines 579-583) previously included the following: “We first studied a diversity dataset of 568 SPB⁻ genomes, 446 of which were generated specifically for this study, 109 had already been published^{27,28,29,31,48,49,50}, and the other 13 were unpublished but deposited in EnteroBase (<https://enterobase.warwick.ac.uk/species/index/senterica>) or GenBank (<https://www.ncbi.nlm.nih.gov/genbank/>) (Supplementary Data 1).”

In Supplementary Data 1 (the list of 568 genomes), the column (BH) entitled “Source” can be used to identify the 28 strains from Connor et al.

As we used 109 published genomes from seven different papers, we did not want to overload the text by providing the numbers of strains used per published paper. Instead, we provide all the necessary details in Supplementary Table 1. However, if the editor would prefer all this information to be provided in the main text, we are willing to comply with this request.

- Please provide percentages for cases where quinolone resistance conferring mutations were documented (lines 294-298).

Authors: This has been done.

- The important observation of azithromycin resistance presented in Figure 5 deserves to be mentioned in the text with the mechanism of resistance explained.

Authors: This has been done: “One isolate (83282), acquired in South America in 2014, contained the *mph(A)* gene encoding a macrolide 2’ phosphotransferase, a common determinant of azithromycin resistance (however, the antimicrobial susceptibility pattern of this isolate was unavailable).”

The term ‘azithromycin’ (i.e. an individual antibiotic name) in Figure 5 can be replaced with ‘macrolides’ (antibiotic class name) to be consistent with other terms in the legend.

Authors: This has been done.

- Table 2 – Suggested heading for the second column – “Years of MRCA (95% HPD)”.

Authors: This has been changed to “Time (year) of the MRCA”.

- Some minor editorial suggestions:

o Rephrase the statement in the Abstract “We show that this pathogen existed in the 13th century” to “Our comparative genomics findings suggest that this pathogen has existed since the 13th century”.

Authors: This has been done.

o Change “exploitation’ to ‘exploration’ (line 391), ‘their colonial and immigration histories’ (line 431) to ‘migration and travel patterns over recent centuries’

Authors: This has been done.

o Change to ‘Universities’ (line 544).

Authors: This has been done.

Reviewer #3 (Remarks to the Author):

This is an interesting and very detailed study of the lineage of Salmonella that is the causative agent of Paratyphi B. It is well written, and I enjoyed reading it. It is certainly worthy of publication and will be of broad interest.

Novelty

I think the paper has broad appeal, both within the field and in related fields. The work itself is original, and also builds well on the existing literature. As well as bringing together and drawing a line under findings from other studies (including those by the authors and others), the paper includes some novel/important findings. Firstly, the detailed historical collection shows strong evidence of phylogeographic clustering, which is not something that has been reported before, and is of epidemiological value and scientific value. Secondly, the pattern of resistance is also interesting, including evidence for increasing resistance in SPB isolates recently. Thirdly, The work around the prophages within SPB is also novel, and the pangenome analysis is also more comprehensive than previous work, due to a combination of the larger historic dataset and the use of long read technologies. Lastly, the observation around S. Onarimon, in the supplementary is also of interest to those who are enthusiasts of the Salmonellae, and it was an unexpected nugget that was also new and interesting.

Support for claims

Overall, the paper has a large amount of detail, and represents a very in-depth genomic study of this interesting pathogen.

The paper brings together historical work very well along with a historical and contemporary genomic dataset to provide the highest resolution exploration to date of SPB. The approaches used are appropriate and well matched to the aims, and the methods, both laboratory and bioinformatics are explained in sufficient detail to enable reproducibility. The use of Enterobase is welcome and enhances the reproducibility of the work. The release of the probes for Mykrobe is also welcome as this is open-source software.

Overall, the data analysis is sensible, and the methods are appropriate and meet the

expected standards in the field.

Other questions/comments for the authors:

Although I like this paper a lot and think it should be published, there are a number of questions/clarifications that would be helpful, these are;

The paper reports the correlation with geography and the evidence of geographic clustering; however, I couldn't see a formal analysis for the phylogeography. Given that the authors had made use of BEAST, I wondered if they had made use of any of the phylogeographic models to examine predicted origins and phylogeographic spread more formally.

So

1. Were analyses performed to substantiate the suggestion of export from Europe?

Authors: We initially performed two phylogeographic reconstructions with [phytools](https://peerj.com/articles/16505/) (<https://peerj.com/articles/16505/>) and PastML (<https://academic.oup.com/mbe/article/36/9/2069/5498561>) to document the exportation of SPB PG1 from Europe at different points.

Phylogeographic reconstruction with PastML

Phylogeographic reconstruction with Phytools

However, due to a deeper sampling for European historical isolates, we were not comfortable presenting these phylogeographic reconstructions, which might be perceived as a formal proof. Instead, we simply suggest that export from Europe may have occurred, based on historical data and the tree topology.

2. The detail around the BEAST analysis is in the supplementary, and these results are interesting, although broadly agreeing with what had been found before - but the BEAST date/context information isn't that well referenced in the main text - so could the authors consider how to better signpost to the detail in the supplementary information regarding BEAST.

Authors: We have now moved all the supplementary data dealing with the Beast analysis to the main text (discussion section). It now reads *“Based on our dataset of 568 SPB⁻ PG1 genomes, we estimated the age of this pathogen at ~750 years (1274 CE; 95% CI, 915 – 1583), which is very close to the previous median date of origin estimated by Connor and coworkers²⁷ (1188 CE; 95% CI, 469 BC – 1799 CE), who used only 25 SPB⁻ PG1 genomes (i.e., those with a known year of isolation). SPB⁻ is older than SPA, which is estimated to have originated 450 – 700 years ago²⁵. SPA was discovered two years after SPB (in the USA in 1898)^{1,2,32} but is currently the most frequent agent of paratyphoid fever²⁸. Due to lineage extinction, in particular, times to the MRCA are often underestimated and the inclusion of ancient DNA in the analysis would increase precision and make it possible to establish dates of origin further in the past²⁵. The dating of a representative collection of modern isolates of SPC estimated the origin of SPC to 456 – 664 years ago. When a draft SPC genome from an 800-year-old Norwegian skeleton was added to the analysis, the time to the MRCA increased to 1162 – 1526 years²⁶. Unfortunately, no ancient DNA is currently available for SPB⁻ strains.”*

The AMR analysis is interesting, but it wasn't clear to me from the paper in what context the resistance genes were found. So;

3. Were the genes found on chromosomal mobile elements, carried with/on phage or were they plasmid carried?

Authors: Most of the AMR detected was due to single point mutations of chromosomal genes (*gyrA* and *gyrB*). The five remaining AMR isolates (those with ESBL, carbapenemase or *mphA* genes in particular) had AMR plasmids. This is now indicated in the main text: *“Between 1898 and 2000, only one isolate (0.3%, 1/345) had antibiotic resistance genes (ARGs). This human isolate (B73-1117), collected in France in 1973, displayed resistance to ampicillin (*bla*_{TEM-1D}), streptomycin (*strAB*, *aadA1*, and *aadA2b*), sulfonamides (*sul1*), chloramphenicol (*cmIA1*), and tetracycline (*tetA*). Between 2001 and 2021, 23.1% (52/223) of isolates had ARGs. One isolate acquired in Turkey in 2001 (01-7995) produced a CTX-M-3 extended-spectrum beta-lactamase³⁰, whereas another isolate (P7704) acquired in South America in 2019 produced an OXA-48 carbapenemase³¹. One isolate (83282), acquired in South America in 2014, contained the *mph(A)* gene encoding a macrolide 2' phosphotransferase, a common determinant of azithromycin resistance (however, the antimicrobial susceptibility pattern of this isolate was unavailable). All these rare isolates contained AMR plasmids (**Supplementary Data 1**); however, the mechanisms of resistance (21.5%, 48/223) most prevalent during the 2001-2021 period involved mutations of the quinolone resistance-determining regions of the chromosomal *gyrA* and *gyrB* DNA gyrase genes leading to resistance to nalidixic acid and/or decreased susceptibility or resistance to ciprofloxacin (minimum inhibitory concentration [MIC] between 0.125 and 0.5 mg/L).”*

4. From the figure in the supplementary, it looks like there weren't many plasmids in the samples analysed, though it would be good to have this described in a little more detail. The

paper mentions a phylogenetic comparison between plasmid genes and core genome, but it isn't clear to me what this means, or what it shows - can the authors explain what is meant and consider if figure 9 in the supplementary is appropriate to communicate this.

Authors: Plasmid-borne AMR was indeed very rare, with only five isolates displaying such resistance. We now provide more detail about the plasmid versus chromosome locations of the AMR genes in the main text (please see above our answer to point 3). The 242 plasmid genes (Supplementary Data 4 and Supplementary Fig. 9) were identified in the pangenome analysis, which was based principally on complete genomes. However, due to the large proportion of short-read sequences in our study, we are not comfortable with the idea of linking the AMR determinants to these 242 plasmid genes. We therefore prefer not to link Supplementary Data 4 and Fig. 9 to the section dedicated to AMR. However, we now indicate in the main text (results section) that these plasmid genes were found in plasmids with and without AMR genes. It reads: “Of the 1,506 accessory genes present in < 95% of the genomes, 696 (46.2%), 242 (16.1%), and 28 (1.9%) were found to belong to prophages, plasmids (with or without AMR genes), and transposases, respectively (**Supplementary Data 4**).”

With respect to the phylogenetics and cgMLST analyses;

5. How many (if any) HC400_1620 samples in Enterobase did not contain the STM 336 SNV? It isn't wholly clear from the text if HC400_1620 is just SPB⁻ strains, or if it includes both SPB⁺ and SPB⁻ strains.

Authors: All the HC400_1620 genomes contained the STM 3356 *d*-Tar⁻ specific SNV.

In the main text (lines 175-181) of our first submission we stated: “We ensured that this diversity dataset comprised exclusively of genomes (i) with the correct *in silico* serotype, (ii) containing the specific SNV associated with the loss of *d*-Tar fermentation in SPB⁻ (ref.²²), (iii) and belonging to the invasive lineage, PG1 (ref.²⁷). This was achieved in a straightforward manner using the HC400_1620 cluster of the Enterobase *Salmonella* core-genome MLST scheme (<https://enterobase.warwick.ac.uk/species/index/senterica>) as a proxy (**Supplementary Note “Validation of the SPB⁻ PG1 diversity dataset”**, **Supplementary Fig. 1**).”

In the Supplementary Note “Validation of the SPB⁻ PG1 diversity dataset” of our first submission we stated (lines 82-87): “All 446 genomes from SPB⁻ isolates and strains contributed by various reference laboratories across the world for this study, and the 109 previously published genomes belonged to HC400_1620 and contained the *d*-Tar⁻ specific SNV (**Supplementary Data 1**).” and “Furthermore, only the HC400_1620 genomes contained the specific SNV within STM 3356 described in SPB⁻ strains¹⁵.”

In the Supplementary Data 1 of our first submission, the column (R) entitled “STM 3356 *d*-Tar⁻ specific SNV” showed that all 568 PG1 genomes contain this SNV.

7. Did the Onarimon samples cluster in a single lineage? were they grouped with other more contemporary samples in a single lineage? it isn't clear from the paper/results where these sit with respect to other lineages within SPB⁻ samples.

Authors: As this serovar was extremely rare, with only one isolate (the serotype reference

strain) in our collection (which includes more than 300,000 clinical isolates and all serovar and variant reference strains of *Salmonella*), it is discussed in the Supplementary Note entitled “Validation of the SPB⁻ PG1 diversity dataset” (lines 82-87): “During this search, we unexpectedly found within HC400_1620, a reference strain (116K) of an extremely rare serotype, Onarimon (antigenic formula: 1,9,12:b:1,2), deposited independently by one of the participating laboratories (Institut Pasteur). Serotype prediction based on genomic sequence confirmed this serotype and the d Tar⁻ specific SNV was present. There were only 28 serotype Onarimon strains reported in 1965 (among the 547,386 strains from diverse sources across the world)¹⁷ and this serotype has been reported to cause paratyphoid fevers¹⁸. As in *Salmonella* spp., serotype antigens can be subject to horizontal gene transfer and homologous recombination¹⁹, we therefore considered 116K to be an O antigen-variant of SPB⁻ and we included it in the study.

We have now modified one sentence and added another to the main text which now reads “Lineage L5 was more frequent in Asia (particularly East Asia), whereas L2 and L9 were more frequently identified in Europe. The only strain (116K) of serotype Onarimon (an O:9 antigen variant of SPB⁻) — isolated in Japan in 1935 — belonged to lineage L5.”.

Lastly;

8. In the validation work around Mykrobe, was the probe tested against a wider database to identify if the probe could miss-detect SPB? what validation was done on this - as the MS only lists testing against the study datasets.

Authors: We have now tested our genomic tool on a genomic dataset of 102 non-SPB PG1 *Salmonella* reference genomes from the SARA, SARB and ATCC collections (new Supplementary Data 10). Interestingly, two rare genotypes (genotypes 7.0 and 7.3) were commonly called (in most *Salmonella* serotypes). The data are presented in detail in the Supplementary Note entitled “Development of a new SNV-based genotyping tool for SPB⁻ PG1”. In the main text (discussion section), we have added a sentence stressing the importance of applying this scheme only to genuine SPB PG1 genomes. It reads “It is, however, important to apply this scheme only to confirmed SPB⁻ PG1 genomes — identified by the cgMLST HC400_1620 cluster or using MLST7 plus the specific d-Tar⁻ SNV — as Mykrobe may otherwise assign the rare genotype 7.0 and to a lesser extent genotype 7.3 to most non-PG1 *Salmonella* genomes (for non-PG1 genomes the tool will, however, always yield “Unknown” instead of “*Salmonella*_Paratyphi B” under the column “species” of the output table).”.

9. In the discussion there are assertions around long term carriage in elderly patients - it wasn't clear to me what the evidence was for this. In those contemporaneous samples from older patients, was there evidence of carriage (e.g., diversity) or long-term chronic infection in their genomes?

Authors: This assertion on long-term carriage in elderly patients in Europe is based on the fact that old SPB PG1 genotypes (those that were epidemic in Europe several decades ago, such as genotype 9.1 in France) are currently found only in elderly patients with no recent history of travel to countries in which PTB is endemic and without enteric fever. Instead, their isolates (some obtained regularly and consistently over time from the same patient) originate from stools (rather than blood samples retrieved during acute bloodstream

infections). We previously wrote *“Interestingly, old genotypes are still being isolated in Europe. For example, 4% of the surveillance isolates from the UK belonged to genotypes 2.1 and 5, and 9.5% of those in France belonged to the 9.1_France genotype. The eight French cases, for which isolates were not recovered from blood samples, were patients between 81 and 98 years of age, suggesting that they may be long-term carriers infected several decades ago.”*. This has been changed to: *“Interestingly, old genotypes are still being isolated in Europe. For example, 4% of the surveillance isolates from the UK belonged to genotypes 2.1 (n = 7) and 5 (n = 1), and 9.5% (n = 8) of those in France belonged to the 9.1_France genotype. The eight French cases, for which isolates were not recovered from blood samples, were patients between 81 and 98 years of age with no recent history of travel to countries in which PTB is endemic, suggesting that they may be long-term carriers infected several decades ago.”*.

10. Were the 'carriage' isolates resistant to antibiotics (which one might have expected - as one would have expected some antibiotic exposure in older patients at some point, which you might have expected to impact carriage of SPB)?

Authors: This would make a lot of sense. However, due to the design of our microbiological study, with the inclusion of very limited metadata, it was impossible to determine whether all our isolates originated from long-term carriers or other types of patient. We cannot, therefore, present a formal comparison of AMR data between long-term carriers and other patients. Three of the eight French potential long-term carriers of genotype 9.1 identified in the surveillance dataset (please see point 9) had *gyrA* mutations (three different types). We now indicate in the main text (discussion section): *“Long-term SPB⁻ PG1 carriers may have particularly high levels of exposure; for example, three of the eight (37.5%) genotype 9.1 isolates recently obtained from elderly French patients (see above) had *gyrA* mutations (of three different types).”*

REVIEWERS' COMMENTS

Reviewer #1 (Remarks to the Author):

The authors have satisfactorily addressed all comments. No further suggestions.

REVIEWERS' COMMENTS

Reviewer #1 (Remarks to the Author):

The authors have satisfactorily addressed all comments. No further suggestions.

Authors: We would like to thank again the three reviewers for their many insightful comments.